# Self-propelling vesicles define glycolysis as the minimal energy machinery for neuronal transport

María-Victoria Hinckelmann[1,2,3,4,†], Amandine Virlogeux[1,2,3,4,5,6,*], Christian Niehage[7,*], Christel Poujol[8,9], Daniel Choquet[8,9], Bernard Hoflack[7], Diana Zala[1,2,3,†] & Frédéric Saudou[1,2,3,5,6,10]

The glycolytic enzyme glyceraldehyde-3-phosphate dehydrogenase (GAPDH) facilitates fast axonal transport in neurons. However, given that GAPDH does not produce ATP, it is unclear whether glycolysis *per se* is sufficient to propel vesicles. Although many proteins regulating transport have been identified, the molecular composition of transported vesicles in neurons has yet to be fully elucidated. Here we selectively enrich motile vesicles and perform quantitative proteomic analysis. In addition to the expected molecular motors and vesicular proteins, we find an enrichment of all the glycolytic enzymes. Using biochemical approaches and super-resolution microscopy, we observe that most glycolytic enzymes are selectively associated with vesicles and facilitate transport of vesicles in neurons. Finally, we provide evidence that mouse brain vesicles produce ATP from ADP and glucose, and display movement in a reconstituted *in vitro* transport assay of native vesicles. We conclude that transport of vesicles along microtubules can be autonomous.

[1] Institut Curie, F-91405 Orsay, France. [2] CNRS, UMR3306, F-91405 Orsay, France. [3] Inserm, U1005, F-91405 Orsay, France. [4] Faculté de Médecine, Univ. Paris Sud11, F-94276 Le Kremlin-Bicêtre, France. [5] Grenoble Institut des Neurosciences, GIN, Univ. Grenoble Alpes, F-38000 Grenoble, France. [6] Inserm, U1216, F-38000 Grenoble, France. [7] Biotechnology Center, Technische Universität Dresden, D-01307 Dresden, Germany. [8] CNRS, UMR 5297, F-33000 Bordeaux, France. [9] Interdisciplinary Institute for Neuroscience, IINS, Univ. Bordeaux, F-33077 Bordeaux, France. [10] CHU Grenoble Alpes, F-38000 Grenoble, France. † Present addresses: IGBMC, CNRS UMR 7104—Inserm U964, F-67404 Illkirch-Graffenstaden, France (M.-V.H.); ESPCI-ParisTech, PSL Research University, F-75005 Paris, France and CNRS, UMR8249, F-75005 Paris, France (D.Z.). * These authors contributed equally to this work. Correspondence and requests for materials should be addressed to F.S. (email: frederic.saudou@inserm.fr).

Fast axonal transport (FAT) is a very efficient mode of delivery in neurons that is mediated by the ATPases kinesin and dynein[1]. It is characterized by high velocity and processivity over long distances. However, the regulatory mechanisms leading to this particularly efficient transport are not clear. A greater number of kinesins may lead to a substantially higher velocity of cargos[2], although this remains uncertain[3]. The organization of the motors themselves on the cargo may also affect the velocity[4]. Emerging evidence indicates that co-factors may play an important role in increasing the efficiency of the motors, as observed for dynein with the Bicaudal D family adaptor protein[5]. We recently reported that in addition to these mechanisms, the glycolytic enzyme glyceraldehyde-3-phosphate dehydrogenase (GAPDH) localized on vesicles may promote efficient FAT by optimizing the energy supply to the molecular motors[6]. However, how GAPDH provides energy to the motors remain to be elucidated. Indeed, although pharmacological or genetic manipulation of GAPDH decreased FAT, GAPDH itself does not produce ATP. Rather, ATP is produced by the downstream glycolytic enzymes phosphoglycerate kinase (PGK) and pyruvate kinase (PK). We found that enriched vesicular fractions can produce ATP when incubated with the substrates of GAPDH, and that increasing the stoichiometric load of GAPDH on vesicles increased FAT; however, we could not exclude the role of additional factors such as NADH, which is produced by GAPDH, or diffusion within the cytoplasm, in providing the substrates necessary for ATP production. An intriguing observation was the presence of PGK in vesicular fractions raising the possibility that in addition to functional GAPDH, the entire glycolytic pathway or at least the pay-off phase of glycolysis may be associated with the vesicles[6]. Given that the entire glycolytic enzyme complex is associated with the plasma membrane of red blood cells[7] and the locally produced ATP fuels $Na^+/K^+$ and $Ca^{2+}$ pumps[8], this metabolic organization may be present on other membranes such as vesicles, even though these are motile. In support, some glycolytic enzymes have been identified in vesicular compartments such as endosomes[9] and synaptic vesicles[10,11] where they could locally provide ATP for the processivity of the ATPase $H^+$ pump[12,13].

Here, using a combination of label-free quantitative proteomics on neuronal motile vesicles and functional transport assays in neurons and in vitro, we determine the minimal energy-producing machinery that is sufficient to propel vesicles in neurons.

## Results

### Purification of motile vesicles.
Many studies have investigated the proteome of endosomes, synaptic or secretory vesicles, leading to an exhaustive list of proteins present in these various organelles[9,14–21]. However, the molecular composition of vesicles that are transported in neurons has not been defined. Specifically, the luminal constituents, transmembrane proteins and associated complexes of dynamic or motile vesicles remain to be elucidated. The identification of these components could improve the understanding of the mechanisms regulating transported vesicles in neurons. To obtain a highly enriched fraction of motile vesicles, we modified a previously described protocol[6] using Thy1:p50-GFP transgenic mice. These mice express the dynactin subunit, dynamitin, fused to green fluorescent protein (GFP) under the neuronal promoter Thy1, at levels below the threshold that disrupts the dynactin complex[22]. Vesicles associated to p50-GFP can thus be considered as motile vesicles as they are likely to associate with the molecular motors.

Mouse forebrains were lysed and subjected to subcellular fractionation, to obtain the 100,000 g fraction or P3 fraction,

known to be enriched in small vesicles and devoid from mitochondria (Supplementary Fig. 1). To selectively enrich for motile vesicles, we performed a magnetic immunopurification (IP) against GFP, thereby selecting specifically small vesicles associated with the molecular motors (Fig. 1a). This motile vesicle fraction (VF; Fig. 1b, IP-GFP lane) was enriched for p50-GFP and for endogenous p50. This is expected, as GFP-dynamitin is incorporated into dynactin at a ratio of 2.2 labelled subunits to 4 total dynamitin subunits per complex[23]. It also contained the dynactin subunit p150$^{Glued}$, which is also a component of the dynactin complex. Both anterograde and retrograde molecular motors were associated with this fraction, as shown by the presence of kinesin heavy chain and dynein intermediate chain (DIC). We controlled our purification protocol by analysing the presence of proteins known to be transported along microtubules (MTs) in axons and dendrites. As expected, we found in the IP-GFP fraction vesicular markers such as synaptophysin that is a well-known integral membrane protein of synaptic vesicles[14], SNAP25 that is transported by FAT to presynaptic membranes[24], the luminal vesicular protein pro-brain-derived neurotrophic factor (BDNF)[25] and the pro-hormone convertase furin that are present in large, dense core vesicles[26], as well as huntingtin, a regulator of axonal transport known to be associated to motile vesicles[27]. Importantly, none of these components were significantly detected in the control IP (Fig. 1b, IP-Control lane), showing the specificity of the immunoisolation. The cytoplasmic protein tubulin was specifically associated, although in low quantity, to the motile VF. This association is expected, as tubulin is known to interact with molecular motors as well as with the dynactin complex.

### Protein composition of motile vesicles.
The protein composition of the purified motile vesicles and of the cytosolic fraction from the same animals was further analysed by liquid chromatography tandem mass spectrometry (LC–MS/MS) after separation by SDS–PAGE. A total of 1,291 proteins were identified as constituents of motile vesicles (Supplementary Data 1). Given the membranous nature of vesicles, we verified the enrichment of the motile fraction by calculating the percentage of proteins that contain transmembrane domains and thus can be associated to membranes. We therefore ranked the proteins that were identified in the motile vesicles fraction (Supplementary Data 1) and in the cytosolic fraction (Supplementary Data 2) by intensity and grouped them into bins of 100 proteins. The number of proteins classified within the Gene Ontology term 'transmembrane proteins' was then recorded for each bin and for the motile vesicles fraction and the cytosolic fraction separately. Transmembrane proteins were highly represented in the high-intensity bins of the motile vesicles fraction (Fig. 1c), especially when compared with the cytosolic fraction. Enrichment of transmembrane proteins decreased for the vesicular fraction and increased for the cytosolic fraction along with the decrease of intensity of the proteins. This demonstrated that our motile VF was enriched in transmembrane proteins, a known feature of vesicles. Consistent with the motile nature of these purified vesicles, among the most abundant components were members of the dynein complex including dynein heavy chain, DIC and dynein light chains, as well as subunits of the dynein-associated dynactin complex (Fig. 1d and Supplementary Data 3). Although less abundant in the sample, both the conventional anterograde motor KIF5C and the neuronal motor responsible for synaptic vesicle precursor anterograde trafficking, KIF1A, were identified (Fig. 1d and Supplementary Data 3). The variety of the different vesicular compartments associated with the molecular motor complex was reflected by the diversity of the different small

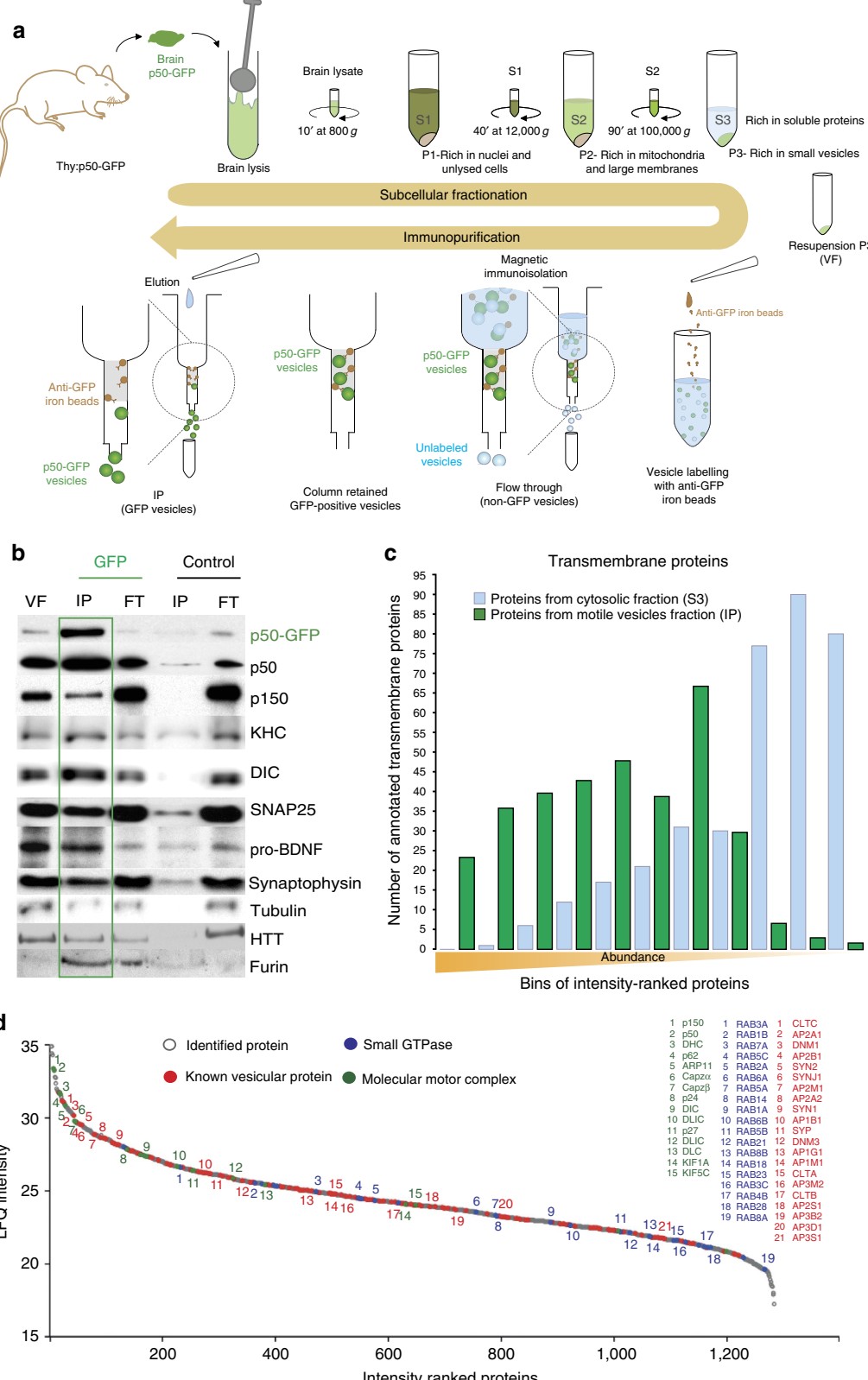

**Figure 1 | Enrichment of motile vesicles.** (**a**) Isolation of motile vesicles. *Thy1:p50-GFP* brains were gently homogenized. The homogenate was subjected to sequential centrifugation steps, to obtain the high-speed pellet containing small vesicles (P3). Motile vesicles were isolated from non-motile vesicles through magnetic IP against GFP. (**b**) The motile VF (IP-GFP) is enriched for transgenic and endogenous p50 compared with the control IP with no primary antibody. It contains anterograde and retrograde molecular motors, as well as vesicular markers. (**c**) Motile vesicles and cytoplasm composition were studied by LC–MS/MS. Comparison of these two fractions shows that motile vesicles are enriched for transmembrane proteins. (**d**) Motile VF identified proteins were ranked by intensity and plotted according to their relative abundance (grey spots). Molecular motors and associated proteins (green spots) are among the most abundant proteins. Numerous components previously identified as vesicular residents are represented in the motile fraction (red spots). Different small GTPases were also found to be associated to the motile vesicles (blue spots).

GTPases and associated proteins (Supplementary Data 4). These included RAB3A and C, known to be associated to synaptic vesicles, several small GTPases associated with endoplasmic reticulum–Golgi vesicular trafficking (RAB1B, RAB2A, RAB6A and B, RAB8A and B, and RAB14), endolysosomal trafficking (RAB7A), early endosomes (RAB5A and C) and endosomes (RAB4B, RAB18, RAB21, RAB5B, C and A, RAB1A and RAB7) and cilia transport (RAB23 and RAB28). In line with these findings, a large variety of proteins associated with the endocytic compartment were identified, including clathrins, dynamin, synaptojanin and components of the adaptor protein complex (Supplementary Data 5). The purified fraction also contained synaptic vesicle residents such as synapsin and synaptophysin, among many other markers of different vesicular compartments. Finally, a large number of proteins not previously reported to be part of vesicles were also detected. These included signalling proteins including phosphatases and kinases, components of the cytoskeleton, chaperones, ribosomal proteins, proteasomal components and some metabolic proteins. These findings suggest that a large set of proteins (the 'transporteome') are present on vesicles that could play a role in regulating vesicular trafficking.

**The glycolytic machinery is associated to motile vesicles.** Our previous observation that PGK is present with GAPDH on vesicles[6] led us to focus on glycolytic enzymes. Surprisingly, in addition to GAPDH and PGK, we detected, in the proteomic analysis of enriched motile vesicles fraction, the ten enzymes of the glycolytic machinery that are required to break glucose down into pyruvate (Supplementary Fig. 2a). Glycolytic proteins were among the most abundant proteins within the proteins found in the motile VF (Fig. 2a, Supplementary Data 6 and Supplementary Fig. 2b). In support, some glycolytic enzymes have also been identified in other proteomic studies on axonal transport-associated fractions[9,28,29]. Usually considered as mere contaminants, some of the glycolytic enzymes might associate to specific structures, to efficiently provide the ATP needed for certain functions[6,30]. We further validated the specificity of the association of glycolytic enzymes to motile vesicles by performing immunoprecipitation experiments followed by western blot analysis. We were not able to detect triosephosphate isomerase due to poor specificity of the antibody. Except phosphoglucose isomerase (PGI) that could not be detected in the IP, we found the eight other enzymes of the glycolytic cascade to be selectively associated to the motile fraction as compared with control immunoprecipitation, where their levels are much lower (Fig. 2b and Supplementary Fig.2b).

We next immunostained all the glycolytic enzymes on rat primary cortical neurons that were grown in microfluidic devices, to separate axons from cell bodies and dendrites[31,32], allowing us to specifically assess their localization in axons (Fig. 2c). To eliminate the cytosolic free portion of proteins, cells were permeabilized before fixation, ensuring that only the proteins associated to cytoskeletal structures or organelles were present. This approach is essential for proteins such as glycolytic enzymes that are abundant in the cytosol. We then analysed the subcellular distribution of hexokinase (HK), PGI, phosphofructokinase (PFK), aldolase (ALDO), GAPDH, PGK, phosphoglycerate mutase (PGM), enolase (ENO) and PK. We first used SNAP25 as a marker for vesicles. Although SNAP25 is a t-SNARE mostly associated to pre-synaptic membrane, it is transported at high velocity[24,33] and is detected as vesicles within axons after permeabilization (Fig. 2d). We first verified the best conditions of use and specificity of the antibodies, and analysed their distribution in axons using total internal reflection fluorescence (TIRF) microscopy (Fig. 2d, upper part of the panels). As for the

immunoprecipitation experiments (Fig. 2b), PGI could not be detected on vesicles and the triosephosphate isomerase antibodies that we tested were nonspecific. Using this approach, we observed co-localization with vesicles of the eight glycolytic enzymes previously detected by biochemical approach (Fig. 2b and Supplementary Fig. 2b). To unequivocally demonstrate this association, we next switched the microscope to super-resolution mode and analysed the distribution of the same fields by ground-state depletion (GSD) microscopy. We observed significant co-localization between endogenous glycolytic enzymes and SNAP25-positive vesicles (Fig. 2d, lower part of the panels).

Although SNAP25 is transported within vesicles, it is not *per se* a canonical marker of any particular type of vesicle. We therefore extended our co-localization experiments with a variety of cargos and glycolytic enzymes in axons of primary cortical neurons grown in microfluidics as described previously (Fig. 2). Given the high number of co-localization, we used Airyscan confocal imaging that provides high resolution with short temporal acquisitions and treatments compared with GSD microscopy (Fig. 3). We first analysed the localization of endogenous glycolytic enzymes with endogenous synaptophysin. Except for HK that showed no co-localization with synaptophysin, the remaining seven glycolytic enzymes for which specific stainings could be obtained, showed strong co-localization (Fig. 3a). We next focused on four glycolytic enzymes including one of the preparatory phase (ALDO) and three of the pay-off phase including the two ATP-producing enzymes PGK and PK. We confirmed the co-localization of glycolytic enzymes with synaptic vesicles precursors as shown by the immunostaining of these endogenous enzymes and exogenously expressed VAMP2-mCherry (Fig. 3b). We next used markers of secretory vesicles including endogenous Chromogranin A and exogenously expressed BDNF-mCherry. Except for PK that showed little co-localization with BDNF-mCherry, most of these enzymes were found on secretory vesicles (Fig. 3c,d). Finally, we investigated the presence of glycolytic enzymes on APP vesicles whose transport is reduced on silencing of the glycolytic enzyme GAPDH[6]. As for synaptic and secretory vesicles, we found the four enzymes to co-localize with APP-containing vesicles (Fig. 3e). To rule out any nonspecific co-localization, we used Ctip2, a nuclear marker of neurons, and found no obvious co-localization, neither with PK immunopositive vesicles nor with synaptophysin immunopositive vesicles (Fig. 3f).

Interestingly, we found that SNAP25-containing vesicles strongly co-localize with synaptophysin (Fig. 3a), suggesting that both cargos are transported within the same vesicles (that is, synaptic precursors vesicles). On the other hand, secretory and synaptic precursor vesicles, although they both contain glycolytic enzymes, showed no overlap, indicating they have different molecular identities but probably depend on glycolytic enzymes for their transport (Fig. 3d). Finally, we found that the presence of glycolytic enzymes is not restricted to vesicles transported anterogradely within axons but also to postsynaptic endosomes within dendrites as shown by the co-localization of PGK with the transferrin receptor in dendrites of cortical axons (Fig. 3g).

Together, using complementary biochemical approach, high and super-resolution microscopies, our findings indicate that most of the glycolytic enzymes, in particular all the enzymes of the ATP-producing pay-off phase, are selectively localized on synaptic precursors and secretory vesicles, as well as endosomes, within axons and dendrites. These findings are in good agreement with previous functional trafficking experiments showing reduced transport of cargos such as synaptotagmin, APP, BDNF and TrkB after GAPDH silencing[6].

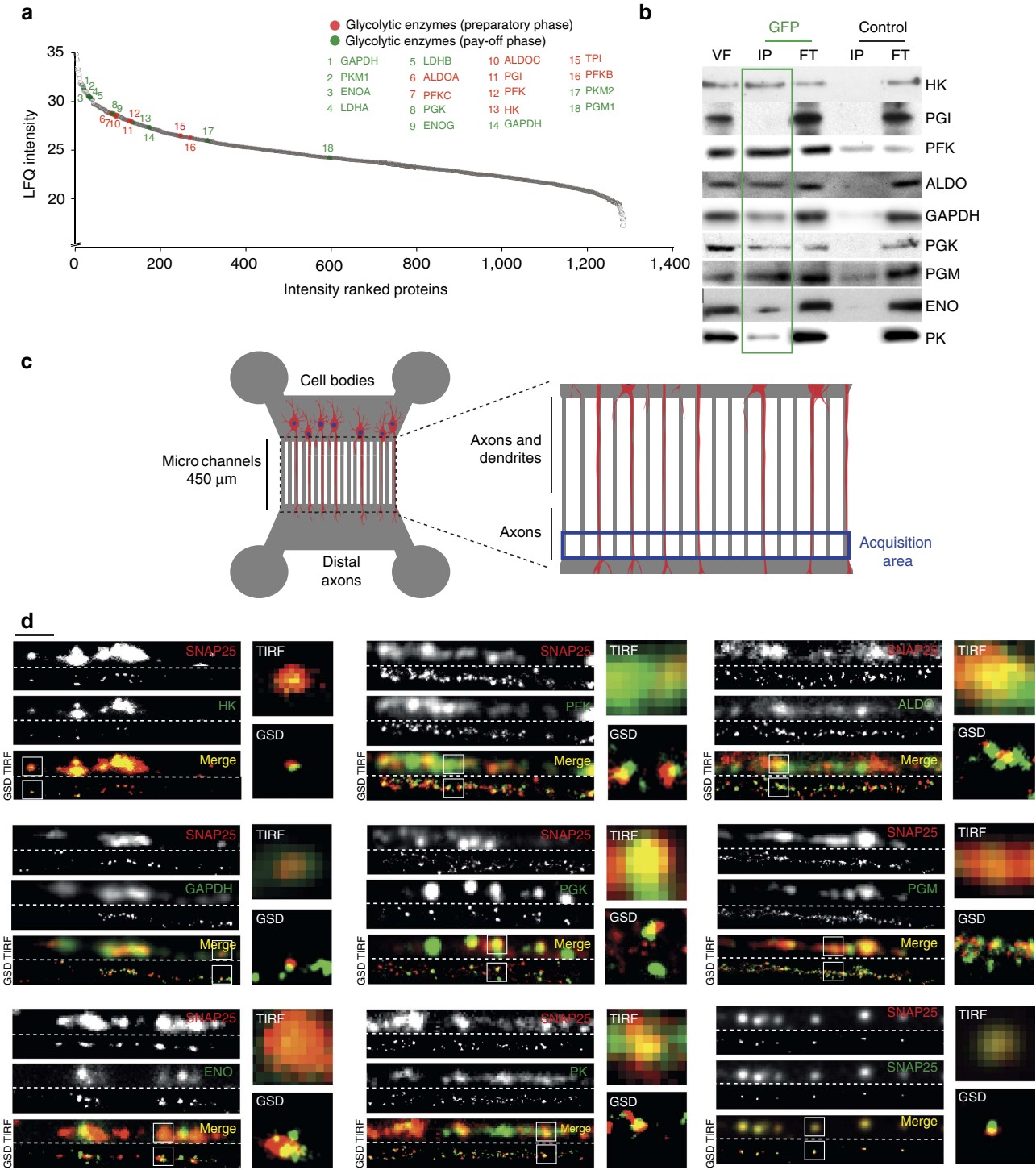

**Figure 2 | Glycolytic enzymes are associated to motile vesicles in axons. (a)** The glycolytic enzymes from both the preparatory (red dots) and pay-off phases (green dots) are present in the motile vesicles and are among the most abundant proteins. **(b)** The presence of the glycolytic machinery is specific to the motile VF, as shown by the absence of these enzymes in the control IP. **(c)** Microfluidic chambers were used to physically separate axons from dendrites and cell bodies. The acquisition area (in blue) was distal from the chamber containing the cell bodies and did not contain dendrites. **(d)** Glycolytic enzymes co-localize with fast-moving vesicles in axons. SNAP25 staining (red channel) co-localize with different glycolytic enzymes (green channel). Co-staining was analysed by TIRF microscopy (upper panels) and by super-resolution (GSD) microscopy (lower panels). Scale bar, 5 μm.

**FAT relies on ATP-producing glycolytic enzymes**. We recently showed that ATP production through the activation of the glycolytic enzyme GAPDH, and not mitochondrial ATP, facilitates FAT[6]. However, GAPDH does not directly produce ATP; ATP is generated by the glycolytic steps involving PGK and PK (Supplementary Fig. 2a). Our findings in the current study

that most of the glycolytic enzymes of the preparatory phase and all of the pay-off phase (Supplementary Fig. 2b) are associated to motile vesicles therefore suggest that a functional glycolytic cascade produces local ATP to promote axonal transport. To test this hypothesis, we used microfluidic devices previously described (Fig. 2c) to specifically assess transport in axons[6,32]. Embryonic

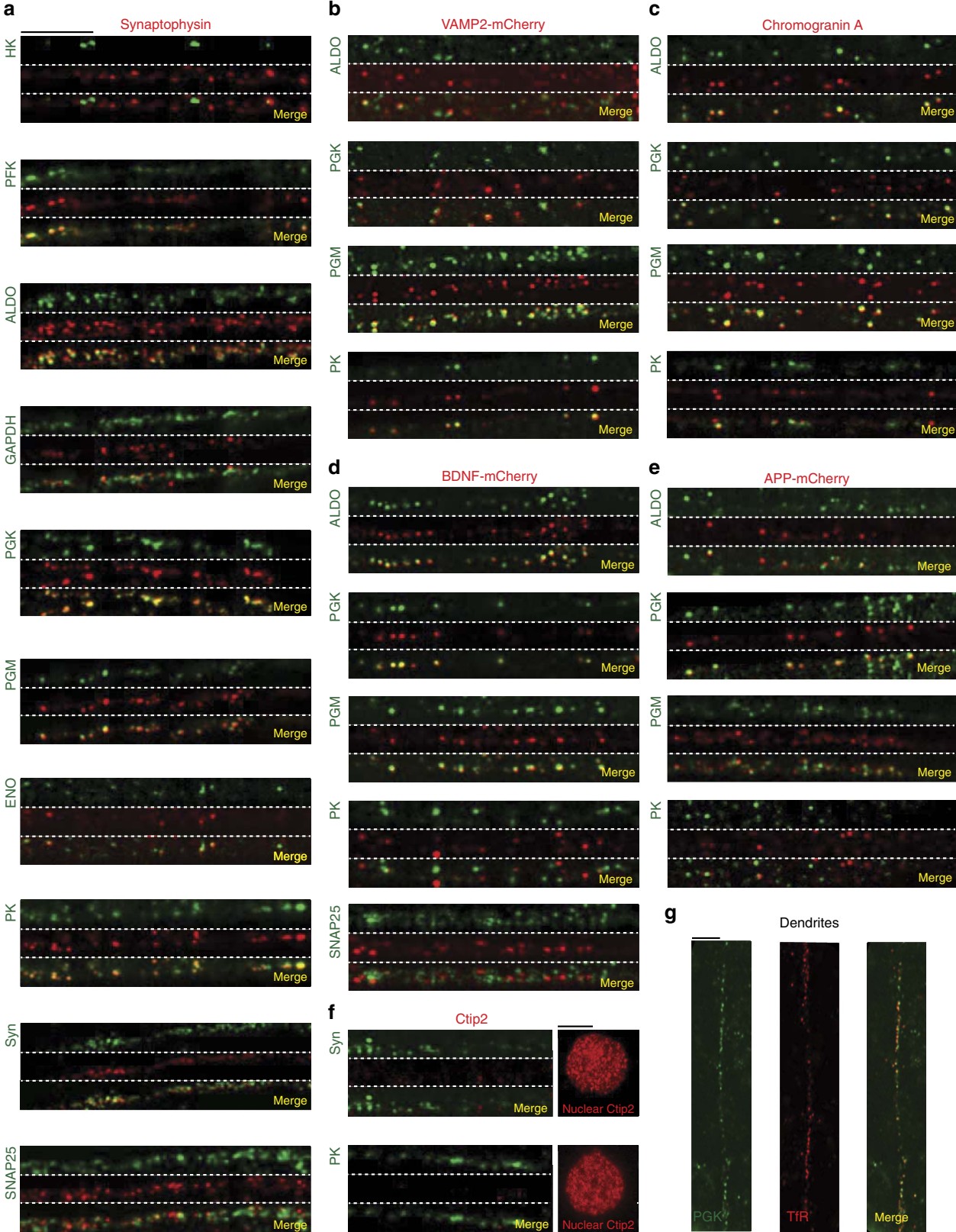

**Figure 3 | Glycolytic enzymes are present on various vesicles in axons and dendrites.** (**a**) Axonal localization of synaptophysin-immunopositive vesicles (red channel) with glycolytic enzymes and SNAP25 (green channel). (**b**) Axonal localization of glycolytic enzymes (green channel) with VAMP2-mCherry-containing vesicles (red channel). (**c,d**) Axonal localization of glycolytic enzymes or SNAP25 (green channel) with secretory vesicles such as chromogranin A-immunopositive vesicles (red channel) (**c**) or BDNF-mCherry-containing vesicles (red channel) (**d**). (**e**) Vesicles expressing APP-mCherry (red channel) show co-localization with glycolytic enzymes (green channel). (**f**) The nuclear neuronal protein Ctip2 (red channel) does not co-localize with synaptophysin of PK (green channel). Nuclear staining of Ctip2 was used as positive control. (**g**) The Transferrin receptor (TfR, red channel) co-localizes with PGK (green channel) in dendrites of cortical neurons. Co-staining was analysed by Airyscan microscopy. Scale bar, 5 μm.

cortical neurons were transfected with BDNF-mCherry, a well-described FAT cargo that is sensitive to GAPDH depletion[6], which we copurified with motile vesicles (Fig. 1b). Transfected neurons were plated in the microchambers and BDNF-mCherry-positive axons were imaged by time-lapse videomicroscopy. We focused first on the pay-off phase, as it is the phase that produces ATP. We therefore systematically silenced all the different enzymes of this phase and verified the efficiency of silencing by immunoblotting experiments (Supplementary Fig. 3a–e). To ensure efficacy of this approach, we selectively targeted the neuronal isoforms of PGK, PGM, ENO and PK, which we identified by MS approach. As we previously reported, GAPDH silencing decreases significantly both anterograde and retrograde velocities (Fig. 4a and Supplementary Movie 1). We found that silencing of the other enzymes, namely PGK, PGM, ENO and PK, significantly reduced the velocity of anterograde- and retrograde-moving vesicles, indicating that the pay-off phase is crucial to sustain efficient trafficking in neurons (Fig. 4a, Supplementary Fig. 3f and Supplementary Movies 2–5). We next tested the contribution of the preparatory phase and tested small interfering RNAs (siRNAs) targeting HK, PGI, PFK and ALDO. However, we observed that these siRNAs were toxic after 24 h in primary neurons, making their use incompatible with FAT assay in microfluidic chambers. To avoid neuronal toxicity and off-target effects, we therefore tested a low dose of 2-deoxyglucose (2-DG), a widely used inhibitor of HK[34], in a medium devoid of glucose and pyruvate. We observed that a 45 min treatment with 25 mM of 2-DG significantly reduced the speed of vesicles (Fig. 4b and Supplementary Movie 6). We then washed out 2-DG and supplemented the medium with 25 mM glucose. This efficiently rescued transport back to control values, demonstrating the requirement of the preparatory phase for efficient FAT. Finally, to determine whether a functional glycolytic cascade is required for transport, we tested whether reduction in vesicle dynamics induced by the silencing of GAPDH can be rescued by addition of phospho-enol pyruvate (PEP), the substrate for PK. We confirmed that reducing levels of GAPDH affects BDNF vesicle velocity (Fig. 4c). However, when this step was bypassed by the addition of PK substrate, velocities returned to normal values (Fig. 4c and Supplementary Movie 7). Together, our results indicate that a functional glycolytic machinery promotes efficient vesicular transport in neurons.

**Native motile vesicles produce glycolytic ATP**. As we found that glycolytic enzymes are bound to motile vesicles, we postulated that purified motile vesicles should be able to generate ATP. To test this hypothesis, purified vesicles (Fig. 1) were incubated with each of the substrates for the different steps of the pay-off phase separately, ADP and cofactors at 37 °C, and ATP production was measured. These purified organelles rapidly generated substantial amounts of ATP (Fig. 5a). Conversely, vesicles without the addition of substrates were unable to produce ATP, showing that they do not carry a reservoir of glycolytic substrates or free ATP; indeed, the levels of ATP were similar to those of the substrates incubated without vesicles. Interestingly, the amount of ATP generated by adding the substrates for the first two steps (GAPDH and PGK) of the pay-off phase of glycolysis produced substantially higher amounts of ATP than the addition of substrates of later steps. This suggests that on vesicles, the pay-off phase functions as a concatenated series of enzymatic reactions, with a higher ATP production when the whole pay-off phase is activated. Given our observation that in addition to pay-off phase enzymes, glycolytic enzymes from the preparatory phase are also present on vesicles, we investigated whether vesicles could

produce significant ATP levels from glucose. Vesicles incubated with 1 mM ADP + Pi, 2 mM NAD + and glucose did not produce ATP. As the preparatory phase requires ATP to initiate the reaction chain, we added limited amounts of ATP (10 μM) that when alone with glucose do not significantly rise ATP levels. Strikingly, we observed a strong production of ATP in these conditions (Fig. 5b). Together, our findings demonstrate that motile vesicles contain a functional glycolytic machinery capable of generating ATP from glucose.

**Native motile vesicles contain functional molecular motors**. We next questioned whether molecular motors present on our purified motile vesicles are functional by measuring their ability to hydrolyse ATP in a MT-dependent assay. On incubation of purified vesicles with polymerized MTs, the amount of ATP decreased over time (Fig. 5c). ATP hydrolysis did not occur in the absence of MTs (Fig. 5c), indicating that this was dependent on MT-associated motor proteins and not by ATPases present on the vesicles. This is consistent with previous findings that molecular motor ATPases require MTs to function[35]. This suggests that purified native motile vesicles contain molecular motors that retain their functionality in transport even after several steps of enrichment.

**Native motile vesicles can self-propel in vitro**. To unequivocally demonstrate that ATP production by the vesicular glycolytic machinery is able to self-propel vesicles in an environment devoid of any cytosolic contaminants and bulk ATP diffusion, we set up a minimal in vitro motility assay composed of brain motile vesicles, MTs and glycolytic substrates. We built a flow chamber with a volume of ~10 μl consisting of a silanized coverslip adhered with double-sided tape to a glass slide. Rhodamine-labelled MTs polymerized in vitro were attached to the coverslip using anti-tubulin antibodies. Finally, purified motile vesicles from mouse brains were fluorescently labelled with a lipophylic dye and loaded into the chamber (Fig. 6a). We then used TIRF microscopy to investigate whether glycolytically produced ATP can sustain vesicular MT-based transport in cytosolic-free conditions. Native vesicles did not show processive movement along MTs in the absence of exogenously added ATP (Fig. 6b). A very small fraction (4%) of MT-bound vesicles did move on MT, but only for short distances (1 μm), probably due to a diffusive movement (Fig. 6f,h). However, the vesicles bound stably to MTs, consistent with dynein association to MTs in the non-nucleotide-bound state (Fig. 6b). As previously reported[23], the addition of 1 mM ATP resulted in the movement of dynamic vesicles along MTs (Fig. 6c). Kymograph analysis of the trajectories showed that these vesicles maintained their association with MTs and moved bidirectionally (Fig. 6f–i). This is consistent with the fact that vesicles need ATP to move. Furthermore, this movement was almost completely blocked when adding vanadate, an ATPase inhibitor that blocks both kinesin and dynein activity, to the flow chamber (Fig. 6d,f–i). Finally, we tested whether the production of glycolytic ATP is sufficient to sustain vesicular transport. We focused on the last step of the glycolytic cascade by incubating native motile vesicles with ADP and PEP, the substrate of the PK reaction. We found that in conditions where ATP is absent but replaced by purified ADP (Supplementary Fig. 4) and the substrates of PK, vesicles were able to self-propel in a processive manner, as observed in the presence of 1 mM ATP (Fig. 6e). The percentage of moving vesicles was similar to that of vesicles incubated with ATP (Fig. 6f). In addition, most of the vesicles were able to move bidirectionally, showing that the ATP generated by vesicles is able to provide energy to both anterograde and retrograde motors (Fig. 6i). The velocities

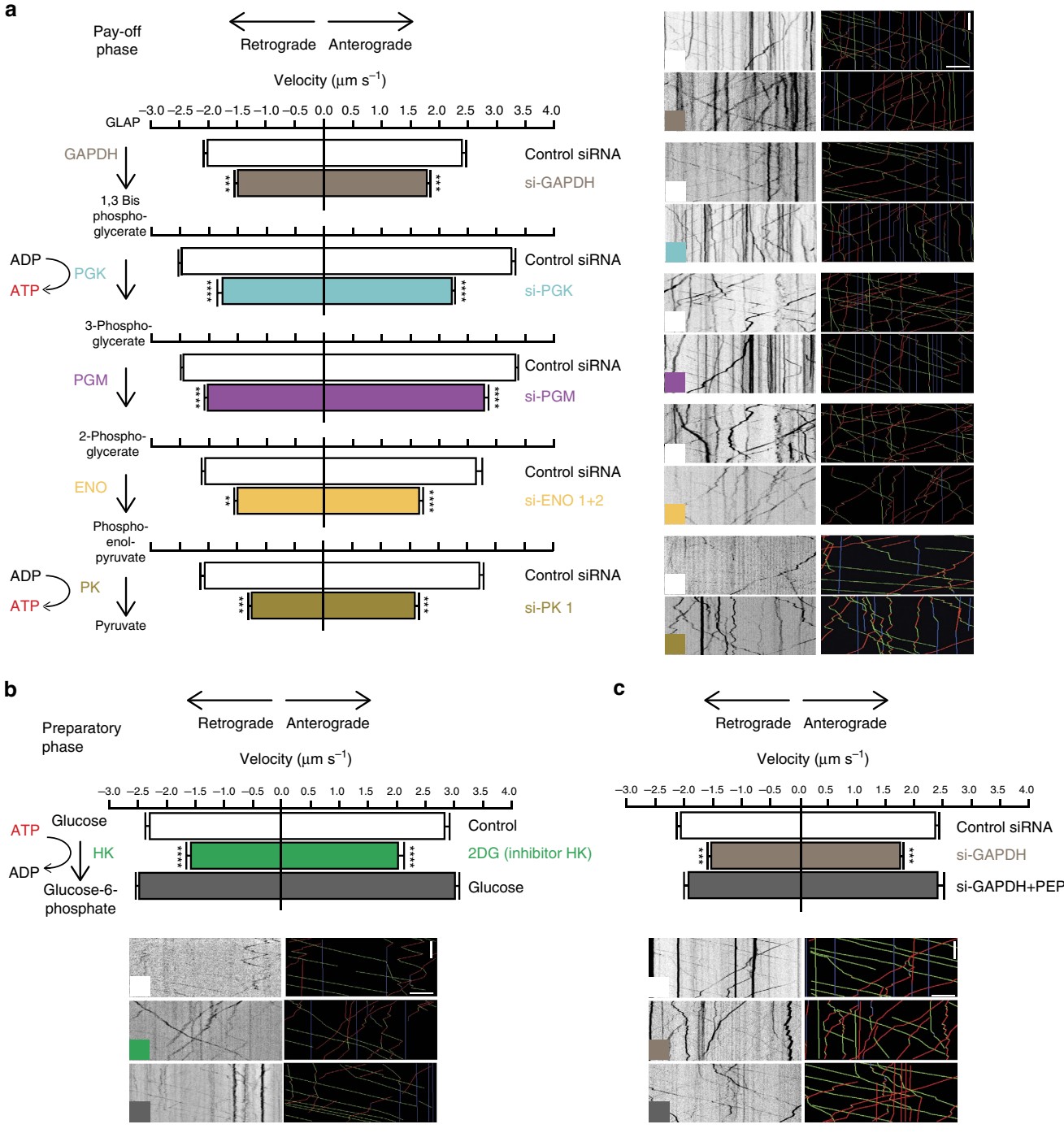

**Figure 4 | FAT relies on glycolytic enzymes. (a)** Silencing of the glycolytic enzymes from the pay-off phase reduces the velocity of BDNF-mCherry vesicles (left panel). Mean and s.e.m. for anterograde and retrograde velocities of control and si-GAPDH (anterograde velocity, $t = 6.355$, $P = 4.3 \times 10^{-10}$, control: $n = 321$, si-GAPDH: $n = 235$; retrograde velocity, $t = 5.4$, $P = 9.8 \times 10^{-8}$, control: $n = 254$, si-GAPDH: $n = 239$), control and si-PGK (anterograde velocity, $t = 11.88$, $P = 1.9 \times 10^{-31}$, control: $n = 973$, si-PGK: $n = 915$; retrograde velocity, $t = 9.6$, $1.8 \times 10^{-21}$, control: $n = 818$, si-PGK: $n = 829$); control and si-PGM (anterograde velocity, $t = 6.289$, $P = 4.2 \times 10^{-10}$, control: $n = 890$, si-PGM: $n = 1{,}126$; retrograde velocity, $t = 5.940$, $P = 3.6 \times 10^{-9}$, control: $n = 802$, si-PGM: $n = 981$); control and si-ENO1 + 2 (anterograde velocity, $t = 6.852$, $P = 1.5 \times 10^{-11}$, control: $n = 455$, si-ENO1 + 2: $n = 325$; retrograde velocity, $t = 6.498$, $P = 1.6 \times 10^{-10}$, control $n = 446$, si- ENO1 + 2 = 335), and control and si-PK1 (anterograde velocity, $t = 9.974$, $P = 2.9 \times 10^{-22}$, control: $n = 441$, si-PK1: $n = 437$; retrograde velocity, control: $t = 11.58$, $P = 7.3 \times 10^{-24}$, $n = 576$, si-PK1: $n = 618$). Representative kymographs showing the trajectories of BDNF-mCherry vesicles of different conditions and analysed trajectories with colour-code red for retrograde, green for anterograde and blue for static vesicles (right panel). Scale bar, 10 μm and 10 s. **(b)** The preparatory phase of glycolysis is required for FAT. Inhibition of HK by 2-DG reduces FAT that is rescued by addition of glucose. Mean and s.e.m. for anterograde and retrograde velocities of control, 2-DG and glucose (anterograde velocity, $F(2{,}784) = 27.96$, $P = 1.9 \times 10^{-12}$, control: $n = 248$, 2-DG: $n = 151$, glucose: $n = 388$; retrograde velocity, $F(2{,}783) = 35.42$, $P = 1.9 \times 10^{-15}$, control: $n = 242$, 2-DG: $n = 165$, glucose: $n = 379$); **(c)** PK activation by PEP rescues transport defect induced by GAPDH silencing. Anterograde and retrograde velocities are represented as mean + s.e.m. (anterograde velocity, $F(2{,}293) = 40.02$, $P = 4.3 \times 10^{-16}$, control: $n = 115$, si-GAPDH: $n = 93$, si-GAPDH + PEP: $n = 87$; retrograde velocity, $F(2, 262) = 34.64$, $P = 4.5 \times 10^{-14}$, control: $n = 112$, si-GAPDH: $n = 97$, si-GAPDH + PEP: $n = 85$). ***$P < 0.001$ and ****$P < 0.0001$.

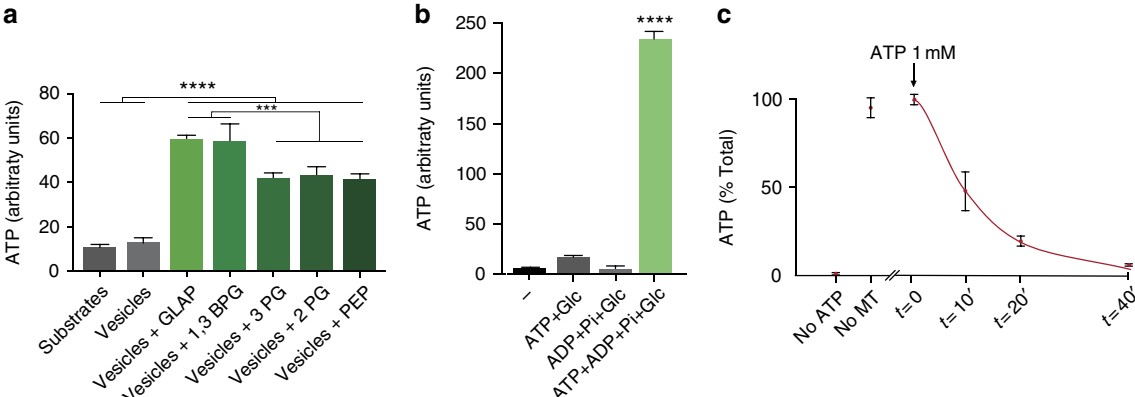

**Figure 5 | Motile vesicles are autonomous energy-producing organelles.** (**a**) The glycolytic pay-off phase is functional on motile vesicles. The graph shows the basal level of luminescence of substrates of the pay-off phase and motile vesicles without substrates. ATP production from purified motile vesicles is observed after incubation with the different substrates of the pay-off phase of the glycolysis. Data are shown as mean and s.e.m. ($F_{(6,14)} = 135.1$, $P = 1.4 \times 10^{-11}$, $n = 3$). (**b**) The vesicular glycolysis is functional and produces ATP from ADP and glucose. Vesicles produce ATP only when incubated with low amounts of ATP to initiate the reaction and ADP + Pi, NAD and glucose. Data are shown as mean and s.e.m. ($F_{(3,8)} = 742.6$, $P = 4 \times 10^{-10}$, $n = 3$). (**c**) Molecular motors retain their kinase activity in purified motile vesicles. Motile vesicles were incubated with polymerized MTs and 1 mM ATP. The ATP reduction overtime in the presence of MTs reflects the presence of active molecular motors. The absence of ATP hydrolysis in the absence of polymerized MTs (No MT) is noteworthy. Data are shown as mean and s.e.m. ($F_{(4,12)} = 155.3$, $P = 9.5 \times 10^{-5}$, $n = 3$). ***$P < 0.001$ and **** = $P < 0.0001$.

observed and the distances covered were also comparable to those of vesicles with exogenously added ATP (Fig. 6g,h). Our findings unequivocally demonstrate that vesicles can move independently of the bulk ATP present in the cytosol and are energetically autonomous due to the presence of a fully functional endogenous glycolytic cascade. Altogether, these results define vesicular glycolysis as the minimal fueling machinery necessary to drive the transport of vesicles on MTs.

## Discussion

In this study, we used an unbiased proteomic approach followed by biochemical methodology and super-resolution imaging, to show that most, if not the whole, glycolytic machinery is present on transported vesicles. Our findings elucidate the mechanism by which GAPDH (which does not produce ATP) enhances FAT in neurons[6]. Specifically, we show that the glycolytic machinery localizes on vesicles and is enzymatically active and fuels the molecular motors for the transport of cargos in axons and in a minimal *in vitro* reconstituted transport assay. The use of native purified vesicles on polymerized MTs unequivocally demonstrates that the embedded glycolytic machinery is sufficient to propel vesicles along MTs without any other sources of energy. The velocity of the movement and the distance travelled on MTs, although much slower than those observed *in vivo*, were consistent with those observed previously in similar experiments[23,36–39]. This is, to our knowledge, the first demonstration of an autonomous motile system for vesicles. However, it does not exclude the possibility that *in vivo*, in axons, other sources of energy such as mitochondrial ATP or lactate from myelin origin[40] could also provide some ATP that may be used by the molecular motors for axonal transport. Nonetheless, this compartmentalization of ATP-producing enzymes coupled closely to molecular motors on moving organelles provides a highly efficient thermodynamic system, as it ensures a locally high ATP concentration that always remains at the proximity of the molecular motors. Interestingly, this compartmentalization emerges as a general mechanism to maintain high local energy concentration in dynamic systems[11,30], not only for ATP but also for GTP, as recently reported for the coupling of nucleoside diphosphate kinase NM23 members to the actomyosin machinery

for contractility at endothelial junctions[41] and for membrane remodelling by acting on dynamin superfamily members[42].

In addition to the whole glycolytic machinery, the label-free quantitative characterization by LC–MS/MS of proteins associated to transported vesicles that we named the transporteome led to the identification of a range of proteins that could be transported on vesicles or small membranes or organelles. Among the 1,291 identified proteins, we detected expected proteins such as components of the molecular motor complex, small GTPases, proteins associated to endocytic vesicles and many transmembrane proteins such as APP and synaptophysin. The number of identified proteins is in the same order of magnitude as, although a little higher than, those identified from previous proteomic analyses on brain clathrin-coated vesicles or synaptic vesicles[17,21,43]. The higher number is probably due to the higher sensitivity of the LC–MS/MS and to the greater diversity of the transported vesicles. Indeed, we found a substantial overlap between synaptic and clathrin-coated vesicle proteins, and an additional number of as yet undescribed vesicular proteins, further supporting our approach.

In conclusion, through this particular strategy aimed at enriching vesicles that are specifically associated to the dynactin complex, we provide an exhaustive list of proteins that are abundant on motile vesicles in neurons and therefore potentially transported over long distances. Importantly, we validate the glycolytic enzymes as being crucial to promote transport in axons and dendrites, and define the minimal components of this cascade as being sufficient to fuel the transport of neuronal vesicles.

## Methods

**Mice.** *Thy1:p50-GFP* transgenic mice expressing low levels of dynamitin fused to GFP were described previously[22]. Animals were maintained with access to food and water *ad libitum* and kept at a constant temperature (19–22 °C) and humidity (40–50%) on a 12:12 h light/dark cycle. Only adult male were used in the study. All experimental procedures were performed in an authorized establishment (Institut Curie, Orsay facility license #C91471108, February 2011) in strict accordance with the recommendations of the European Community (86/609/EEC) and the French National Committee (2010/63) for care and use of laboratory animals under the supervision of authorized investigators (permission #91–448 to S. Humbert). This study was evaluated and approved by the ethics committee, 'Comité d'éthique en matière d'expérimentation animale Paris Centre et Sud' (National registration number: #59) presided by Pascal Bigey.

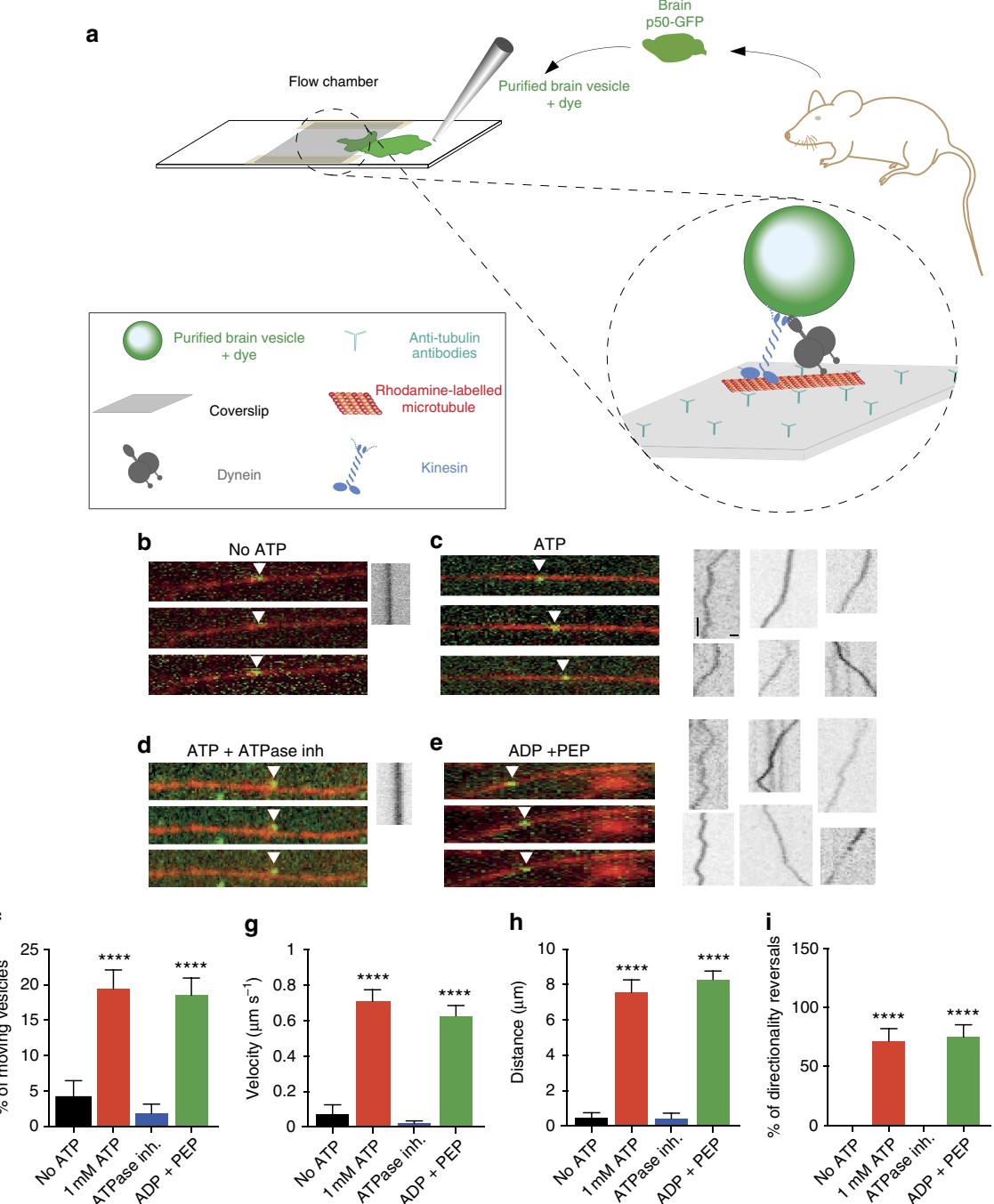

**Figure 6 | Self-propelling vesicles *in vitro*.** (**a**) Representation of native *in vitro* motility test. The flow chamber was assembled using a silanized coverslip and glass slide, spaced by double-sided tape. Anti-tubulin antibodies, blocking solution and rhodamine-labelled MTs were sequentially loaded into the chamber, to ensure specific attachment of MT. Motile vesicles purified from mouse brain and stained with the green fluorescent lipophilic dye DIO were finally incubated in the flow chamber. MTs and motile vesicles were imaged using TIRF microscopy. (**b**) In the absence of ATP, motile vesicles were able to attach to MTs, but were not dynamic. (**c**) The addition of ATP into the chambers resulted in vesicular movement. (**d**) Addition of 100 µM vanadate stopped vesicle motility. (**e**) Vesicles incubated with PEP and ADP were dynamic. (MTs in red, motile vesicles in green, kymograph representing movement right panel). (**f–i**) Different analysed dynamic parameters of native motile vesicles *in vitro*, represented as mean ± s.e.m. (No ATP: $n=108$, ATP 1 mM: $n=120$, ATPase inh: $n=147$, ADP + PEP: $n=117$. ***$P<0.001$ and ****$P<0.0001$. (**f**) $P=1.7\times10^{-82}$, $F(3, 488)=192.9$; (**g**) $P=4.3\times10^{-15}$, $F(3, 458)=591.6$; (**h**) $P=4.2\times10^{-19}$, $F(3, 463)=914.7$; (**i**) $P=1.17\times10^{-42}$, $F(3, 281)=95.57$). Scale bar, 1 µm and 2 s.

**Mouse brain vesicle purification.** Motile vesicles were purified as shown in Fig. 1a. *Thy1:p50-GFP* mouse forebrains were homogenized in lysis buffer (HEPES 4 mM, sucrose 320 mM pH 7.4) containing protease inhibitor cocktail (Sigma-Aldrich). The homogenate was first spun for 10 min at 800 g to remove unlysed cells and nuclei. The supernatant was then centrifuged for 40 min at 12,000 g to pellet mitochondria and large membranes. The resulting supernatant was further centrifuged for 90 min at 100,000 g (Beckman, TLA 100.1). The cytosol-enriched supernatant was kept for MS analysis and the high-speed pellet enriched for small vesicles (VF) was resuspended in lysis buffer. To selectively purify small vesicles bound to the molecular motor complex, we immunopurified dynamitin-GFP-positive vesicles. For this purpose, the VF was incubated with anti-GFP iron µBeads or iron µBeads not bound to antibodies as control (Miltenyi Biotec) for 30 min at 4 °C. Separation µColumns (Miltenyi Biotec) were placed in a magnetic field and the incubated solution applied onto the column. The flow

through containing unbound material was collected and then the column was rinsed three times with lysis buffer. The motile VF (IP) was eluted from the column by applying 150 μl of elution buffer (0.1 M triethylamine pH 11.8, 0.1% Triton X-100) in a tube containing 9 μl of 1 M MES pH 3 for neutralization. ForMS, proteins were precipitated using 10% trichloroacetic acid (TCA) and resuspended in Laemmli buffer. For native *in vitro* motility tests and to keep the complex molecular motor-vesicle-associated proteins intact while removing cytosolic contaminants, the VF was further fractionated by flotation through a sucrose step gradient. Sucrose gradient consisted in steps of 0.6, 1.5 and 2.0 M sucrose that was spun for 2 h at 40,000 g at 4 °C. Then, vesicles were isolated from the 0.6 to 1.5 M interface as previously described[44].

**MS and enrichment analysis.** Samples were fractionated by cutting full lanes after SDS–PAGE into ten slices. Proteins in each slice were reduced, carbamidomethylated at the cystein residues and in-gel digested by trypsin in a 1:50 ratio. Tryptic peptides were separated on a 15 cm reversed-phase column (C18, 15 cm × 75 μm, 2 μm beads of 100-Å pore size; Dionex) within a 60 min gradient from 0.1% formic acid/5% acetonitrile to 0.1% formic acid/40% acetonitrile operated by an Ultimate 3000 HPLC system (Dionex). MS was performed with a LTQ Orbitrap XL mass spectrometer (Thermo Scientific), equipped with a nanoESI source (Proxeon). The top eight peaks in the mass spectra (Orbitrap; resolution, 60,000) were selected for fragmentation (CID; normalized collision energy, 35%; activation time, 30 ms, q-value, 0.25). Dynamic exclusion was enabled (repeat count, 2; repeat duration, 10 s; exclusion duration, 20 s). MS/MS spectra were acquired in the LTQ in centroid mode. Proteins were identified using the MaxQuant software package version 1.2.2.5 (MPI for Biochemistry, Germany) and UniProt database version 04/2013. Carbamidomethylation of cysteine was chosen as a fixed modification; acetylation of the amino terminus, deamidation of asparagine and oxidation of methionine as variable modifications. Proteins with at least one unique and two razor peptides were kept when present in at least two of three replicates. An in-house R-script (www.r-project.org) was used to average protein intensities within sample types, rank proteins in order of decreasing intensity and count the number of proteins associated with particular Gene Ontologies or criteria within bins of 100 proteins.

**Microfluidic chambers and neuronal plating.** Microfluidic devices were generated as described previously[6]. Embryonic rat cortical neurons (E17) were electroporated with BDNF-mCherry using Amaxa Nucleofactor kit (Lonza). Neurons were then plated in the proximal chamber in Neurobasal supplemented with 2% B27, 1% GlutaMAX and 1% penicillin/streptomycin. Videomicroscopy and immunolabelling acquisitions were done in the distal part of the microgrooves 3–5 DIV, when axons have already crossed the microchannels (Fig. 2c).

**Live-cell imaging and analysis.** *In vivo* axonal transport of BDNF-mCherry was imaged in cortical neurons plated in microfluidic chambers. Neuronal processes can grow into the microchannels, but only axons can reach the distal part of the micro channel located 450 μm away from the proximal chamber where cell bodies are located (Fig. 2c). Acquisitions were done in the distal part of the microchannels, using an inverted microscope (Axio Observer, Zeiss) with a × 63 1.46 numerical aperture (NA) oil-immersion objective coupled to a spinning-disk confocal system (CSU-W1-T3; Yokogawa) connected to an electron-multiplying CCD (charge-coupled device) camera (ProEM+ 1024, Princeton Instrument) and maintained at 37 °C and 5% CO2. Images were taken every 0.2 s for 30 s. Kymographs and velocity analysis were done using KymoToolBox[6], a homemade plugin for ImageJ (NIH, USA). Briefly, calibrated kymographs were generated from maximal projection of time-lapse videos. Vesicle trajectories were then analysed manually, to extract the dynamic parameters of the tracked particles. Only moving vesicles were analysed; therefore, static vesicles or particles moving at velocities lower than 0.2 μm s$^{-1}$ were not taken into account.

**Constructs drugs and siRNA.** BDNF-mCherry construct has been previously published[45] and has been used for axonal transport studies. Lentiviruses expressing BDNF-mCherry and APP-mCherry were used for immunostaining experiments. Briefly, BDNF-mCherry and APP-mChery[46] were cloned in a pSin vector for lentivirus production. Rat PFK A, B and C, ALDO A and C, PGK1, PGM1, and ENO1 and 2 were silenced using Mission pre-designed siRNA (Sigma-Aldrich). Rat GAPDH was silenced with a previously published siRNAII[6]. Rat PK was silenced with annealed siRNA I 5′-CAU-CUC-CCU-GCA-GGU-GAA-GGA-GAT-TT and UCU-CCU-UCA-CCU-GCA-GGG-AGA-UGT-TT-3′ (Eurogentec), siRNA II 5′-GAU-UUU-GGA-GGC-CAG-CGA-UGG-AAT-TT-3′ and 5′-UUC-CAU-CGC-UGG-CCU-CCA-AAA-UCT-TT-3′ (Eurogentec), or siRNA III 5′-GAG-CCU-CCA-GUC-AAU-CCA-CAG-AC5-5-3′ and 5′-GUC-UGU-GGA-UUG-ACU-GGA-GGC-UC5-5-3′ (Eurogentec). The control siRNA used is a universal negative siRNA (OR-0030-neg05; Eurogentec). siRNAs were electroporated and plated on microfluidic chambers and on 35 mm dishes. For each time-lapse videomicroscopy experiments, immunoblotting of primary cortical neurons were performed in parallel, to validate the downregulation of the studied glycolytic enzyme. Neurons were analysed for FAT and protein levels 4 to 5 days after plating. Only the experiments in which a significant silencing of the glycolytic

enzyme was observed were quantified for axonal transport. For the HK inhibition, Neurobasal B27 medium was replaced by a buffer without glucose and pyruvate (HEPES 10 mM, NaCl 145 mM, KCl 5.5 mM, MgCl2 1.2 mM, CaCl2 1.1 mM pH = 7.4) but containing 25 mM 2-DG.

**Native *in vitro* motility test.** Rhodamine-labelled MTs were generated by incubating for 20 min at 37 °C unlabelled tubulin purified from porcine brain and rhodamine-labelled tubulin (cytoskeleton) in BRB80 buffer (PIPES 80 mM, MgCl2 1 mM, EGTA 1 mM) containing 1 mM GTP, to allow polymerization. Polymerized MTs were then stabilized by adding Taxol 50 μM and incubated again for 20 min at 37 °C. Free tubulin was eliminated by centrifuging the polymerized MTs for 10 min at 14,000 g. The pellet was resuspended in BRB80 buffer containing 40 μM Taxol.

Native *in vitro* assays were carried out in a flow chamber made of a slide attached to a silanized coverslip by two-sided tape. Flow chambers were first incubated for 5 min with 2% anti-tubulin antibody (Sigma Aldrich) and then blocked with 5% Pluronic F127 (Sigma-Aldrich). MTs were allowed to bind in the flow chamber for 10 min and the excess was removed by washing with BRB80 100 mM dithiothreitol and 20 μM Taxol. Finally, purified native vesicles, previously operated or not with MgATP or 2 mM PEP and 0.2 mM ADP in motility buffer (BRB80, 0.05% Pluronic F127, 20 μM Taxol, 0.25 mg ml$^{-1}$ BSA, 50 mM dithiothreitol, 0.5 mg ml$^{-1}$ glucose oxidase, 470 U ml$^{-1}$ catalase and 15 mg ml$^{-1}$ glucose) were added to the flow chamber (Fig. 5a). Sodium orthovanadate (100 μM; New England Biolabs) was added in a subset of experiments into the motility buffer.

Acquisitions were made using an inverted microscope (Elipse T*i*, Nikon) with a × 60 1.42 NA APO TIRF oil-immersion objective (Nikon) coupled to a CCD camera (CoolSnap, Photometrics) and maintained at 37 °C and 5% CO2. A first picture of rhodamine-labelled MTs was taken and vesicular movement was recorded by acquiring images every 0.2 s. Kymographs and dynamic parameters were obtained using KymoToolBox.

**ATP production and consumption analysis.** To test molecular motor activity after vesicular purification, 10 μl of 'motile vesicles' were incubated with 1 mg of polymerized MTs and 1 mM MgATP at 30 °C. Aliquots (10 μl) were removed at different time points and the reaction was stopped by storing them at 4 °C. To measure ATP content, 2 μl of Cell Titer-Glo Luminescent Viability Assay (Promega) was added and luminescence was recorded.

To assess ATP production by the different enzymes of the pay-off phase of glycolysis that specifically associate with 'motile vesicles', 10 μl of this fraction was incubated at 37 °C with the different substrates and co-factors (2 mM GLAP, 3 mM 1,3BPG, 2 mM 3PG, 4 mM 2PG, 2 mM PEP, 0.2 mM ADP, 2 mM NAD and 2 mM Pi) in motility buffer. For ATP production by glucose, 'motile vesicles' were incubated with 500 μM Glucose, 10 μM ATP, 1 mM ADP, 2 mM Pi and 2 mM NAD. ATP content of the samples was measured as described above

**Western blotting analysis.** Protein concentration was assessed using a BCA kit (Pierce). Proteins (10 μg) were denatured at 95 °C for 10 min in loading buffer. Six, 8, 10 12 and 15% acrylamide gels were loaded. Proteins were transferred onto nitrocellulose membranes and the blocked in 5% non-fat milk in TBS buffer, 0.1% Tween. Primary antibodies and secondary antibodies were incubated for 1 h at room temperature. Images of the western blotting experiments have been cropped for presentation. Full-size images are presented in Supplementary Figs 5 and 6.

**Immunolabelling STORM and Airyscan microscopy.** Neurons were fixed 3–5 days after plating in the microchambers. A 2 min prelysis was done using 0.5% saponine in PBS, after which cells were fixed for 20 min using 4% paraformaldehyde and 4% sucrose in PBS. Once fixed, microchambers were rinsed three times with PBS and blocked for 1 h at room temperature with 3% BSA + 0.3% Triton X-100. Primary antibodies were incubated overnight at 4 °C in blocking buffer and secondary antibodies were incubated for 1 h at room temperature.

Super-resolution images were taken in a Leica SR GSD (Leica Microsystems, Mannheim, Germany) GSD microscope, equipped with a HCX PL APO × 100 objective (Leica Microsystems) with a 1.47 NA for TIRF illumination. Images were acquired on a sensitive electron-multiplying CCD iXon3 camera (ANDOR, Belfast, UK). The lasers used were 405, 488, 532 and 642 nm diodes with approximate powers of 30 mW, 300 mW, 1 W and 500 mW, respectively (Coherent, Santa Clara, CA, USA).

High-resolution images were acquired using a Zeiss LSM 710 inverted confocal microscope using Airyscan detector, to improve signal to noise and resolution, and a × 63 objective (1.4 NA, Zeiss).

**Reagents and antibodies.** The following antibodies and dilutions were used for western blotting (WB) and immunofluorescence (IF): mouse anti-GFP horseradish peroxidase (Miltenyi Biotec, 130-091-833, 1:5,000 WB), mouse anti-p150$^{glued}$ (BD Transduction Laboratories, 610474, 1:1,000 WB), mouse anti-kinesin heavy chain (Chemicon, MAB1614, 1:500 WB), mouse anti-DIC (Millipore, MAB1618, 1:500 WB), mouse anti-synaptophysin (Sigma-Aldrich, S5768, 1:1,000 WB, 1:500 IF), rabbit anti-synaptophysin (Abcam, ab14692, 1:500 IF),

mouse anti-α-tubulin (Sigma-Aldrich, T9026, 1:1,000 WB), rabbit anti-BDNF (Chemicon, AB1534, 1:1,000 WB), rabbit anti-p50 dynamitin (Millipore, AB5869P, 1:250 WB), mouse anti-SNAP25 (Abcam, ab24737,1:2,000 WB and IF), rabbit anti-HK I (Cell Signaling, mAb2024, 1:1,000 WB, 1:100 IF), mouse anti PGI (Santa Cruz Biotechnology, sc-30392, 1:500 WB), rabbit anti-PFK (Cell Signaling, #5412P, 1: 1,000 WB, 1:100 IF), goat anti-ALDO (Santa Cruz Biotechnology 1:200 WB, sc-12059, 1:50 IF), rabbit anti-GAPDH (Santa Cruz, sc-32233, 1:1,000 WB, 1:100 IF), mouse anti-PGK1 (Abcam, ab90787, 1:1,000 WB), goat anti-PGK (Santa Cruz, sc-23805, 1:1,000 WB, 1:100 IF), goat anti-PGM (Santa Cruz Biotechnology, sc-67756, 1:200 WB, 1:100 IF), rabbit anti-ENO (Santa Cruz Biotechnology, sc15343, 1:200 WB, 1:100 IF), rabbit anti-ENO1 (Cell Signaling, #3810, 1:1,000 WB), rabbit anti-ENO2 (Cell Signaling, mAb8171, 1:1,000 WB), rabbit anti-PK (Cell Signaling, mAb3190,1:1,000 WB, 1:100 IF), mouse anti-Chromogranin A (Santa Cruz, sc-393941, 1:500 IF), rabbit anti-Furin (Santa Cruz, sc-20801, 1:1,000 WB), rabbit anti-Glud1 (Abcam, EPR11369, 1:1,000 WB), rat anti-Ctip2 (Abcam, ab18465, 1:500 IF), mouse anti-mCherry (Abcam, ab125096, 1:1,000 IF), mouse anti-Transferrin receptor (Institut Curie, A-M-M#32, 1:500 IF) and rabbit anti-LDH (Cell Signaling, mAb3582, 1:1,000 WB). ATP, ADP, 2-DG, GLAP, 1-3BPG, 3PG, 2PG, Glucose and PEP are from Sigma-Aldrich.

**Statistical analysis.** All experiments consisted of at least three independent replicates. Statistical analyses were done using GraphPad Prism (GraphPad Software, Inc.). All data are presented as mean ± s.e.m. Groups were compared using unpaired two-tailed t-test for dual comparisons or one-way analysis of variance followed by Dunnett's post-hoc analysis for multiple comparisons. Data distribution was assumed to be normal. Sample size was not assessed by any statistical method. No randomization was done. Investigator was not blind for data collection or analysis.

**Data availability.** The authors declare that the data supporting the findings of this study are available within the article and its Supplementary Information or from the corresponding author upon request.

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

## Acknowledgements

We thank S. Humbert for support and discussions, and members of the Saudou and Humbert labs for helpful comments and/or reading of the manuscript. We thank Y. Saoudi and the imaging facility platform (PIC-GIN) for help with image acquisitions; C. Benstaali for mouse care; R. Mallik, J. Ross, E. Perlson, C. Leduc, F. Farina, S. Leveque-Fort and D. Loew for technical advices; C. Janke for the purified tubulin; T. Ryan for VAMP2-mCherry; M.P.Teulade-Fichou and M. Bombled for HPLC analysis; G. Froment, D. Nègre and C. Costa from the lentivirus production facility/SFR BioSciences Gerland—Lyon Sud (UMS3444/US8). p50-GFP mice were kindly provided by E. Holzbaur. This work was supported by grants from Agence Nationale pour la Recherche (ANR-12-BLAN-SVSE2-HURIT and ANR-14-CE35-0027-01-PASSAGE, F.S.), Fondation pour la Recherche Médicale (FRM, équipe labellisée) (F.S.), Fondation Bettencourt Schueller (F.S.), the Huntington Society of Canada (F.S) and INSERM (F.S.). We thank Association Huntington France for fellowship support (M.-V.H.).

## Author contributions

M.-V.H. performed, analysed and interpreted the data of biochemical purifications and immunoprecipitations, ATP production and *in vitro* transport assay, and wrote the manuscript. C.N., M.-V.H. and B.H. performed the MS experiments and subsequent analyses. M.-V.H., C.P. and D.C. performed and analysed the super-resolution experiments. M.-V.H and A.V performed and analysed FAT in neurons. A.V. performed and analysed high-resolution experiments and ATP production by glucose. D.Z. interpreted the data, conceived and designed the study, and wrote the manuscript. F.S. conceived and designed experiments, analysed and interpreted the data, conceived and designed the study, and wrote the manuscript. All authors reviewed the manuscript.

## Additional information

**Competing financial interests:** The authors declare no competing financial interests.

