## [Peer Review File · Nature Communications]

Reviewers' comments:

Reviewer #1 (Remarks to the Author):

The authors present here convincing evidence to the point that at least a fraction of brain small organelles (vesicles) are associated to glycolytic enzymes, and that these are used in producing ATP from ADP. They further show elegant evidence to the point that this ATP may be used for propelling the vesicles, for instance in axons, both in vitro and in cell cultures.

I find the manuscript overall solid, and worth publishing. However, I am puzzled by one aspect, namely by the use of SNAP25 as a marker for vesicles. It is well known that the wide majority of SNAP25 molecules in synapses are found on the plasma membrane. As the authors point out, it is transported rapidly down the axons, which suggests it must be found in some vesicles. Nevertheless, SNAP25 is not the most logical choice for a general understanding of axonal vesicles, where one would expect that the authors would test synaptic vesicle or active zone precursor molecules.

SNAP25 is transported in an unusually rapid fashion, so fast that it was termed Superprotein at one point in the past (see Loewy, Liu, Baitinger and Willard, Journal of Neuroscience, 1991). Therefore, it is probably typical for all vesicles. Therefore, I suggest that the authors test at least two more markers in their immunostainings in Figure 2, such as synaptophysin, as a marker of precursors of synaptic vesicles, and a soluble protein which is only loosely associated to vesicles, as a negative control (for example, alpha-synuclein).

Additionally, the investigation of the same type of colocalization should be performed in dendrites, based on, for example, colocalization with the transferrin receptor, as a post-synaptic endosome marker. This would strengthen the phenotype described by the authors both in the pre- and postsynaptic compartments, and would make the manuscript's message clearer.

Reviewer #2 (Remarks to the Author):

Manuscript 94575 - Hinckelmann et al.

The manuscript by Hinckelmann et al. describes that motile transport vesicles in mammalian neurons contain a functional glycolytic enzyme complex for their energy supply. Proteomics of affinity purified biochemical fractions using the dynactin complex identified all glycolytic enzymes in the glycolytic pathway. Colocalization of glycolytic enzymes with motile vesicles was confirmed with superresolution imaging of cultured neurons grown in microfluidic chambers. Pharmacological and genetic interventions on these glycolytic enzymes resulted in reduced velocity of transported vesicles. In cell-free experiments the authors show that the glycolytic enzyme complex produces ATP autonomously. Furthermore, incubation of microtubules with purified motile vesicles, ADP and PEP, a glycolytic substrate, resulted in movement of vesicles along microtubules, indicating that the vesicles autonomously produced ATP that generated force to drive vesicle transport.

These data provide important new insights into the mechanisms of vesicle transport in the brain and probably other tissues. Glycolytic enzymes as the energy source for transport vesicles have been proposed before (Zala et al. Cell 2013), but the current data show for the first time that vesicles can autonomously produce energy for microtubule based transport using an on-board functional glycolytic enzyme complex. The manuscript contains multiple lines of evidence that vesicles have all components present of the glycolytic machinery and produce ATP autonomously. The proteomic analysis of purified fractions provides compelling evidence that the enzyme complex is part of the vesicle/motor complex. The cell free experiments show for the first time that the presence of this enzyme complex is sufficient for vesicle transport. The knock-down and pharmacological interventions show their function is necessary. The quality of the experiments is generally excellent, with several innovative assays

(superresolution imaging microfluidic chambers, cell free assay). These assays are a great asset for follow up research also for transport disorders such as Huntington and other neurodegenerative disorders. The proteomic analysis is important for the identification of new molecules functioning in vesicle transport, which is of broad interest.

Most conclusions in this manuscript are convincing and based on multiple lines of evidence. However, the conclusion that the purified biochemical fractions are motile vesicles is not supported by the data, the evidence for the gene-silencing data in Figure 3 seems over-interpreted and the use of SNAP25 as a vesicle marker is not convincing.

MAJOR ISSUES

1. The claim that purified biochemical fractions are motile vesicles is not justified

The Western blot of a purified biochemical fraction in Fig 1B contains several markers for transport vesicles, but only one cargo protein (pro-BDNF). However, the proteomic analysis of the same fraction does not identify pro-BDNF. Moreover, other cargo molecules are also absent in the fraction that the authors label "motile vesicles". One of the most often used markers for post-Golgi transport vesicles (such as BDNF-vesicles) are the chromogranins. These are not found in the "motile vesicles" fraction, but they are in the S3. Hence, the purification appears to de-enrich for vesicle cargo. Proteolytic enzymes known to localize to BDNF-vesicles (pro-hormone convertases, carboxy peptidase etc.) are also absent from "motile vesicles" fraction. Other neuropeptides (-precursors) known to accumulate in motile vesicles are not found either. It seems likely that the purified fraction is not intact vesicles, but probably a fraction enriched in vesicle membranes. The authors should either provide more conclusive evidence that this fraction is indeed purified vesicles using electronmicroscopy and Western blot/ELISA for other cargo, or rephrase their definition of the purified fraction throughout the manuscript.

2. Silencing of different glycolytic enzymes does not prove the presence of functional glycolytic enzymes on vesicle.

Upon silencing different glycolytic enzymes (Figure 3), the authors conclude (line 246/247): "Together, our results indicate that a functional glycolytic machinery is on board motile vesicles and promotes FAT in axons". However, global silencing of glycolytic enzymes could result in a global disruption of glycolysis in the cell. These data show that the glycolytic machinery supports fast axonal transport, but in itself does not prove that the motile vesicles contain a functional glycolytic machinery on board.

3. SNAP25 as vesicle marker for immunostainings

Although SNAP25 is known to be associated to vesicles, it mainly resides in the plasma membrane most often in defined micro domains, which might be mistaken for vesicles. Therefore, it is not ideal to use SNAP25 as the only marker for motile vesicles in figure 2D. Why were known vesicle markers such as secretogranins/chromogranins, BDNF or VAMPs not used? An additional staining showing co-localization between SNAP25 and another vesicle marker would provide convincing evidence for the use of SNAP25 as a vesicle marker.

Minor issues:

- Figure 2A: number 15 in graph green -> red
- Fig 3 and 4 show no real data, only group averages. It would be better to add some kymographs at least in Fig 3.
- Page 9, line 197 Fig 1b should be Fig 2b?
- It would be informative to add supplemental table with identified proteins known to function in glycolytic machinery (as supplemental table 4 and 5)
- Co-localization analysis SNAP-25 with glycolytic enzymes (Supplemental figure 2): explain method of analysis and indicate N/n numbers
- References to supplementary movies mismatch:
 - o Supplementary movie 2 (line 237) -> movie S6

o Supplementary movie 3 (line 245/246) -> movie SS7

Reviewer #3 (Remarks to the Author):

The manuscript by Hincklmann et al., demonstrates a mechanism for local production of ATP on neuronal vesicles by means of glycolysis, that is sufficient to mediate their transport on microtubules. The authors elegantly showed the association of the glycolytic enzymes in Dynactin-enriched vesicle fractions. Together with the super-resolution based imaging that showed close proximity of axonal vesicles and glycolysis enzymes, the ATP-producing and ATP-independent transport capacity are strongly convincing that a local, vesicle-coupled glycolysis activity can provide sufficient energy for microtubule-based transport.

However, the authors should provide further control experiments data to examine alternative ATP-producing mechanisms, such as lack of mitochondria in their vesicle-purified fraction. Other major concerns:

1. In their examination of the effects of 2-DG HK inhibitor (Fig. 3) the authors show a significant, however not dramatic, ~25% reduction in transport velocity, suggesting that ATP is still available via other sources. The authors should examine whether different time/dosage of glycolysis inhibition will have more robust effect on transport. Also, the involvement of mitochondrial respiration should be examined - if glycolysis is the primary ATP source for the transported vesicles, mitochondrial redox inhibition or decoupling should have a delayed effect in comparison. In this regard, it is crucial to show the relative contribution of both sources of ATP on transport robustness.
2. In Fig 4a, the authors should provide data on ATP production using also preparatory-phase substrates such as Glucose with ADP and low concentration of ATP to initiate the process. That way all the glycolysis process can be validated in this vesicle fraction system.
3. The authors mention that ATP hydrolysis by extracted vesicles did not occur in the absence of MT in-vitro, but should provide the data.
4. In their in-vitro motility assay, the mechanistic dissection of the reported, ATP-independent motility is lacking. It needs to be shown that preparatory phase substrates could fuel motility similar to pay-off phase substrates, and that inhibition of glycolysis abolishes this capacity. Again, mitochondrial involvement should be ruled out by redox/ATPase disrupting drugs.

Minor issues:

5. Details on the specific methodology used to track, filter and analyze the axonal and in-vitro motility is severally lacking as this could greatly influence the outcome of the analysis results.
6. It should be recognized that as most evidence on the glycolysis-machinery association and function is based on dynactin-enriched vesicular fraction from whole brain areas, these are not necessarily long distance, axon-transported specific compartment. Therefore the authors should refrain from emphasizing their conclusions regarding axonal transport specificity of their findings and model.
7. The authors should also refer to other proteomic studies on axonal transport associated fractions (Michaelvski et al., MCP 2010, Debaisieux et al., MCP 2015, Gershoni-Emek et al., MCP 2015), some of which have shown the association of several glycolytic enzymes in their analysis.

Overall, this is a very interesting and well-sustained work, providing a novel insight on the mechanisms of neuronal transport and motor based transport in general. Should the authors address the above comments, I suggest it will be accepted for publication.

Answer to reviewer's comments

Reviewer #1 (Remarks to the Author):

The authors present here convincing evidence to the point that at least a fraction of brain small organelles (vesicles) are associated to glycolytic enzymes, and that these are used in producing ATP from ADP. They further show elegant evidence to the point that this ATP may be used for propelling the vesicles, for instance in axons, both in vitro and in cell cultures.

I find the manuscript overall solid, and worth publishing. However, I am puzzled by one aspect, namely by the use of SNAP25 as a marker for vesicles. It is well known that the wide majority of SNAP25 molecules in synapses are found on the plasma membrane. As the authors point out, it is transported rapidly down the axons, which suggests it must be found in some vesicles. Nevertheless, SNAP25 is not the most logical choice for a general understanding of axonal vesicles, where one would expect that the authors would test synaptic vesicle or active zone precursor molecules.

SNAP25 is transported in an unusually rapid fashion, so fast that it was termed Superprotein at one point in the past (see Loewy, Liu, Baitinger and Willard, Journal of Neuroscience, 1991). Therefore, it is probably typical for all vesicles. Therefore, I suggest that the authors test at least two more markers in their immunostainings in Figure 2, such as synaptophysin, as a marker of precursors of synaptic vesicles, and a soluble protein which is only loosely associated to vesicles, as a negative control (for example, alpha-synuclein).

Additionally, the investigation of the same type of colocalization should be performed in dendrites, based on, for example, colocalization with the transferrin receptor, as a post-synaptic endosome marker. This would strengthen the phenotype described by the authors both in the pre- and postsynaptic compartments, and would make the

manuscript's message clearer.

We thank the reviewer for his appreciation about our study and his positive recommendation.

We acknowledge reviewer's concerns about the vesicular marker that we used. Reviewer 2 and the editor also shared this concern. Our rationale to choose SNAP25 was based on evidence that this protein is transported in axons by FAT to the presynaptic membrane ¹. However, to demonstrate the localization of the glycolytic enzymes to a variety of vesicles including synaptic vesicles (requested by reviewer 1 and 2), secretory vesicles (requested by reviewer 2), we performed additional immunostaining using high-resolution Airyscan confocal microscopy.

As suggested by reviewers 1 & 2, we analyzed co-localization of glycolytic enzymes with endogenous synaptophysin for synaptic vesicles, Chromogranin A to detect secretory vesicles, transferrin receptor to mark post-synaptic endosomes. The latest localization was examined in dendrites. We also extended the characterization of vesicles that co-localizes with glycolytic enzymes to vesicles containing BDNF, VAMP2 or APP. As these requested immunostainings represent a large number of experimental conditions, we performed co-localization experiments of synaptophysin with 7 of the glycolytic enzymes (for which the antibodies are working properly and, selected 4 of the glycolytic enzymes for their co-localization with Chromogranin A, BDNF-mCherry, APP-mCherry and VAMP2-mCherry. The reason to choose Aldolase was based on the fact that this enzyme belongs to the preparatory phase. We selected 3 enzymes from the pay-off phase and in particular PGK and PK as they are the ATP-producing enzymes of the glycolysis. We decided not to repeat immunostainings with GAPDH as we previously characterized its vesicular localization and function in FAT². As shown in new figure 3, except for BDNF-mCherry that showed little co-localization with PK, we observed for all the other

immunostaining, a strong co-localization of the different cargos with the various glycolytic enzymes.

Finally, we also observed co-localization of glycolytic enzymes (in this case PGK) with transferrin receptor in dendrites of cortical neurons suggesting that post-synaptic endosomes are also fueled by glycolysis.

Although reviewer suggested synuclein as a loosely associated protein and because we found synuclein being enriched in the motile vesicle fraction (Table 1) we used Ctip2 as negative control for localization to vesicles and did not observed co-localization neither with PK nor with synaptophysin on vesicles (new Figure 3).

As suggested by reviewer 2, we also validated SNAP25 as a cargo that is present on precursor of synaptic vesicles by performing co-localization experiments of SNAP25 with synaptophysin (new Figure 3).

All these new immunostaining experiments are shown in new Figure 3 and in the corresponding text pages 9-10.

In conclusion, we previously showed that GAPDH was specifically localized on vesicles and that silencing GAPDH decreased the FAT of both BDNF-mCherry, TrkB-containing signaling endosomes, APP-containing vesicles in mouse neurons as well as synaptotagmin-containing vesicles in fly motoneurons. Together with the new experiments provided in this study, we unequivocally demonstrate that most if not all glycolytic enzymes are present on a large number of vesicles including precursors of synaptic vesicles, secretory vesicles and endosomes and that they provide energy for neuronal transport.

Reviewer #2 (Remarks to the Author):

Manuscript 94575 - Hinckelmann et al.

The manuscript by Hinckelmann et al. describes that motile transport vesicles in mammalian neurons contain a functional glycolytic enzyme complex for their energy supply. Proteomics of affinity purified biochemical fractions using the dynactin complex identified all glycolytic enzymes in the glycolytic pathway. Colocaliation of glycolytic enzymes with motile vesicles was confirmed with superresolution imaging of cultured neurons grown in microfluidic chambers. Pharmacological and genetic interventions on these glycolytic enzymes resulted in reduced velocity of transported vesicles. In cell-free experiments the authors show that the glycolytic enzyme complex produces ATP autonomously. Furthermore, incubation of microtubules with purified motile vesicles, ADP and PEP, a glycolytic substrate, resulted in movement of vesicles along microtubules, indicating that the vesicles autonomously produced ATP that generated force to drive vesicle transport.

These data provide important new insights into the mechanisms of vesicle transport in the brain and probably other tissues. Glycolytic enzymes as the energy source for transport vesicles have been proposed before (Zala et al. Cell 2013), but the current data show for the first time that vesicles can autonomously produce energy for microtubule based transport using an on-board functional glycolytic enzyme complex. The manuscript contains multiple lines of evidence that vesicles have all components present of the glycolytic machinery and produce ATP autonomously. The proteomic analysis of purified fractions provides compelling evidence that the enzyme complex is part of the vesicle/motor complex. The cell free experiments show for the first time that the presence of this enzyme complex is sufficient for vesicle transport. The knock-down and pharmacological interventions show their function is necessary. The quality of the experiments is generally excellent, with several innovative assays (superresolution imaging microfluidic chambers, cell free assay). These assays are a

great asset for follow up research also for transport disorders such as Huntington and other neurodegenerative disorders. The proteomic analysis is important for the identification of new molecules functioning in vesicle transport, which is of broad interest.

Most conclusions in this manuscript are convincing and based on multiple lines of evidence. However, the conclusion that the purified biochemical fractions are motile vesicles is not supported by the data, the evidence for the gene-silencing data in Figure 3 seems over-interpreted and the use of SNAP25 as a vesicle marker is not convincing.

We thank reviewer 2 for his comments and constructive suggestions. As recommended, additional work has been performed to address these issues.

MAJOR ISSUES

1. The claim that purified biochemical fractions are motile vesicles is not justified. The Western blot of a purified biochemical fraction in Fig 1B contains several markers for transport vesicles, but only one cargo protein (pro-BDNF). However, the proteomic analysis of the same fraction does not identify pro-BDNF. Moreover, other cargo molecules are also absent in the fraction that the authors label "motile vesicles". One of the most often used markers for post-Golgi transport vesicles (such as BDNF-vesicles) are the chromogranins. These are not found in the "motile vesicles" fraction, but they are in the S3. Hence, the purification appears to de-enrich for vesicle cargo. Proteolytic enzymes known to localize to BDNF-vesicles (pro-hormone convertases, carboxy peptidase etc.) are also absent from "motile vesicles" fraction. Other neuropeptides (-precursors) known to accumulate in motile vesicles are not found either. It seems likely that the purified fraction is not intact vesicles, but probably a fraction enriched in vesicle membranes. The authors should either provide more conclusive evidence that this fraction is indeed purified vesicles using

electronmicroscopy and Western blot/ELISA for other cargo, or rephrase their definition of the purified fraction throughout the manuscript.

We understand reviewer's concerns about the nature of the purified fraction. The fact that certain proteins detailed above were not identified does not reflect their absence. One of the shortcomings of mass spectrometry analysis is that not 100% of the protein components of the samples are efficiently identified (for a review see^{3, 4}. Indeed, the western blots used as representative images for the publication were done with an aliquot of the samples used for the mass spectrometry, therefore although pro-BDNF was not identified by the mass spectrometry analysis, there were present in the purified fraction as we were able to show its presence by immunoblotting. To further demonstrate the enrichment in our preparation for proteins present in motile vesicles, we immunoblotted the purified fraction using antibodies against furin. Furin is known to be a dense core vesicle resident protein. As shown in new figure 1b, furin is enriched in the IP-GFP fraction that is enriched for motile vesicles (e.g.: vesicles that are associated with molecular motors). This new result is described page 6 of the revised manuscript.

Reviewer 2 also questioned the integrity of our vesicle preparation, while we cannot guarantee that the vesicles were intact just before mass spectrometry analysis, we followed the same protocol than the one used for the demonstration that GAPDH is located on vesicles by cryo-tomo-electron microscopy. This localization and the integrity of the vesicle can be visualized on figure 5 of ². To further convince the reviewer, we provide an additional EM image of such vesicles after immunomagnetic isolation. Image on the right shows such vesicles with iron particles that are used for the IP.

2. Silencing of different glycolytic enzymes does not prove the presence of functional glycolytic enzymes on vesicle.

Upon silencing different glycolytic enzymes (Figure 3), the authors conclude (line 246/247): "Together, our results indicate that a functional glycolytic machinery is on board motile vesicles and promotes FAT in axons". However, global silencing of glycolytic enzymes could result in a global disruption of glycolysis in the cell. These data show that the glycolytic machinery supports fast axonal transport, but in itself does not prove that the motile vesicles contain a functional glycolytic machinery on board.

We agree with the overstatement pointed out by the reviewer. Text has been modified and now says: Together, our results indicate that a functional glycolytic machinery promotes efficient vesicular transport in neurons (see revised version page 12).

3. SNAP25 as vesicle marker for immunostainings

Although SNAP25 is known to be associated to vesicles, it mainly resides in the plasma membrane most often in defined micro domains, which might be mistaken for vesicles. Therefore, it is not ideal to use SNAP25 as the only marker for motile vesicles in figure 2D. Why were known vesicle markers such as secretogranins/chromogranins, BDNF or VAMPs not used? An additional staining showing co-localization between SNAP25 and another vesicle marker would provide convincing evidence for the use of SNAP25 as a vesicle marker.

We acknowledge reviewer's concerns about the vesicular marker that we used. Reviewer 1 and the editor also shared this concern. Our rationale to choose SNAP25 was based on evidence that this protein is transported in axons by FAT to the presynaptic membrane¹. However, to demonstrate the localization of the glycolytic enzymes to a variety of vesicles including synaptic vesicles (requested by reviewer 1

and 2), secretory vesicles (requested by reviewer 2), we performed additional immunostaining using high-resolution Airyscan confocal microscopy.

As suggested by reviewers 1 & 2, we analyzed co-localization of glycolytic enzymes with endogenous synaptophysin for synaptic vesicles, Chromogranin A to detect secretory vesicles, transferrin receptor to mark post-synaptic endosomes. The latest localization was examined in dendrites. We also extended the characterization of vesicles that co-localizes with glycolytic enzymes to vesicles containing BDNF, VAMP2 or APP. As these requested immunostainings represent a large number of experimental conditions, we performed co-localization experiments of synaptophysin with 7 of the glycolytic enzymes (for which the antibodies are working properly and, selected 4 of the glycolytic enzymes for their co-localization with Chromogranin A, BDNF-mCherry, APP-mCherry and VAMP2-mCherry. The reason to choose Aldolase was based on the fact that this enzyme belongs to the preparatory phase. We selected 3 enzymes from the pay-off phase and in particular PGK and PK as they are the ATP-producing enzymes of the glycolysis. We decided not to repeat immunostainings with GAPDH as we previously characterized its vesicular localization and function in FAT². As shown in new figure 3, except for BDNF-mCherry that showed little co-localization with PK, we observed for all the other immunostaining, a strong co-localization of the different cargos with the various glycolytic enzymes.

Finally, we also observed co-localization of glycolytic enzymes (in this case PGK) with transferrin receptor in dendrites of cortical neurons suggesting that post-synaptic endosomes are also fueled by glycolysis.

Although reviewer 1 suggested synuclein as a loosely associated protein and because we found synuclein being enriched in the motile vesicle fraction (Table 1) we used Ct1p2 as negative control for localization to vesicles and did not observed co-localization neither with PK nor with synaptophysin on vesicles (new Figure 3).

As suggested by reviewer 2, we also validated SNAP25 as a cargo that is present on precursor of synaptic vesicles by performing co-localization experiments of SNAP25 with synaptophysin (new Figure 3).

All these new immunostaining experiments are shown in new Figure 3 and in the corresponding text pages 9-10.

In conclusion, we previously showed that GAPDH was specifically localized on vesicles and that silencing GAPDH decreased the FAT of both BDNF-mCherry, TrkB-containing signaling endosomes, APP-containing vesicles in mouse neurons as well as synaptotagmin-containing vesicles in fly motoneurons. Together with the new experiments provided in this study, we unequivocally demonstrate that most if not all glycolytic enzymes are present on a large number of vesicles including precursors of synaptic vesicles, secretory vesicles and endosomes and that they provide energy for neuronal transport.

Minor issues:

We thank the reviewer for pointing out these issues. We have modified figures and text accordingly.

- Figure 2A: number 15 in graph green -> red

According to the legend and as pointed by the reviewer, we have now changed the dots corresponding to the enzymes of the preparatory phase in red

Fig 3 and 4 show no real data, only group averages. It would be better to add some kymographs at least in Fig 3.

Figure 3 now contains representative kymographs for all the experimental conditions.

Page 9, line 197 Fig 1b should be Fig 2b?

We have modified the text line 197, page 9 (now line 198).

It would be informative to add supplemental table with identified proteins known to function in glycolytic machinery (as supplemental table 4 and 5)

We added a new table (table 6) with all the identified glycolytic enzymes

- Co-localization analysis SNAP-25 with glycolytic enzymes (Supplemental figure 2): explain method of analysis and indicate N/n numbers

Co-localization of SNAP-25 vesicles with glycolytic enzymes was performed manually. Quantification corresponds to two different neuronal cultures in microchambers and the number of vesicles included in the analysis for each condition are: HK= 128, PFK=163, ALDO=126, GAPDH=230, PGK=221, PGM=293, ENO= 80, PK=189, SNAP25=204. In the new version of the manuscript we further extended this characterization by using new markers of specific types of vesicles including synaptic vesicles precursors, secretory vesicles and endosomes, with several glycolytic enzymes (see new Figure 3). Therefore, to simplify the message and in order to make the manuscript more comprehensible we removed Supplementary figure 2. Although we considered this data important in our previous version of the article, in view of the new information gathered during the revision process we now think that it does not add much in terms of information and the message we want to convey.

- References to supplementary movies mismatch:
 - o Supplementary movie 2 (line 237) -> movie S6
 - o Supplementary movie 3 (line 245/246) -> movie SS7

We thank the reviewer for pointing out these mismatches. These have been corrected.

Reviewer #3 (Remarks to the Author):

The manuscript by Hincklmann et al., demonstrates a mechanism for local production of ATP on neuronal vesicles by means of glycolysis, that is sufficient to mediate their transport on microtubules. The authors elegantly showed the association of the glycolytic enzymes in Dynactin-enriched vesicle fractions. Together with the super-resolution based imaging that showed close proximity of axonal vesicles and glycolysis enzymes, the ATP-producing and ATP-independent transport capacity are strongly convincing that a local, vesicle-coupled glycolysis activity can provide sufficient energy for microtubule-based transport.

However, the authors should provide further control experiments data to examine alternative ATP-producing mechanisms, such as lack of mitochondria in their vesicle-purified fraction.

We understand reviewer's concern about the presence of mitochondria in the enriched fraction. We extensively analyzed the role of mitochondria in our previous publication on the role of GAPDH in FAT of vesicles and found that mitochondria were dispensable for FAT of vesicles². As suggested by reviewer 3, we provide however evidence that mitochondria are unlikely to be present in our motile vesicle fraction based on the observation that the mitochondrial matrix protein Glud1 is absent in the P3 fraction that corresponds to small vesicles fraction and that is the starting point of our immune-magnetic purification using anti-GFP antibody (see figure 1a). This important control experiment is now shown in Supplemental Figure 1 and described in the text page 5.

Other major concerns:

1. In their examination of the effects of 2-DG HK inhibitor (Fig. 3) the authors show a significant, however not dramatic, ~25% reduction in transport velocity, suggesting that ATP is still available via other sources. The authors should examine whether different time/dosage of glycolysis inhibition will have more robust effect on transport.

We choose to present experiments in which 2-DG was used at low concentration and after short incubation time in the microfluidic chambers as we noticed high toxicity when higher concentration and longer incubation time were used. Of course, in these conditions, transport was completely blocked. Because this effect could be due to massive death of neurons, we thought that presenting conditions in which HK is not fully inhibited would avoid potential experimental artifacts. As this was a concern for reviewer 3, we now better explain the experimental conditions (Results-FAT relies on ATP producing glycolytic enzymes- page 11-12)

Also, the involvement of mitochondrial respiration should be examined - if glycolysis is the primary ATP source for the transported vesicles, mitochondrial redox inhibition or decoupling should have a delayed effect in comparison. In this regard, it is crucial to show the relative contribution of both sources of ATP on transport robustness.

We extensively addressed the relative contribution of mitochondria and of glycolysis in the study by Zala and colleagues² (2013). In this study we showed that inhibiting mitochondria ATP synthase by different doses of oligomycin or preventing mitochondria to localize in axons by overexpression of Milton-C had no effect on the fast axonal transport of vesicles in axons. These experiments using both drugs and genetic silencing of GAPDH in neurons and in vivo in fly motoneurons demonstrated that the glycolytic enzyme GAPDH but not mitochondria were necessary for FAT². As we discussed in this Ms, it does not rule out that in some circumstances mitochondria-produced ATP could contribute to FAT. However, we believe that it is not the scope of this study whose goal is to study the role of the whole glycolytic machinery in neurons and in vitro as the minimal machinery capable to self-propel

vesicles.

2. In Fig 4a, the authors should provide data on ATP production using also preparatory-phase substrates such as Glucose with ADP and low concentration of ATP to initiate the process. That way all the glycolysis process can be validated in this vesicle fraction system.

*We thank the reviewer for suggesting this very important but challenging experiment. We used purified vesicles and investigated whether such vesicles could produce significant ATP levels from glucose. When vesicles were incubated with only 1 mM ADP + Pi and glucose they did not produce ATP. As the preparatory phase requires ATP to initiate the reaction chain, we added to Glucose and ADP + Pi and a small amount of ATP (10 μ M). Strikingly, we observed a strong production of ATP in these conditions (**New Fig. 5b**). As a control, when there is only small amount of ATP and glucose but no ADP +Pi, no ATP is produced. Together, our findings demonstrate that motile vesicles contain functional glycolytic machinery capable of generating ATP from glucose.*

3. The authors mention that ATP hydrolysis by extracted vesicles did not occur in the absence of MT in-vitro, but should provide the data.

This information has now been added to figure 5c.

4. In their in-vitro motility assay, the mechanistic dissection of the reported, ATP-independent motility is lacking. It needs to be shown that preparatory phase substrates could fuel motility similar to pay-off phase substrates, and that inhibition of glycolysis abolishes this capacity. Again, mitochondrial involvement should be ruled out by redox/ATPase disrupting drugs.

We agree that this could be important to demonstrate that the preparatory phase is participating to the transport process. However, as pointed out by the reviewer and

as shown in new figure 5b, this requires adding ATP to the motility buffer. According to studies that investigated Hexokinase (HK) in vitro, maximum HK activity is reached when ATP concentration is 2 mM and 1 mM when glucose-6-P is present^{5, 6}. This is compatible with the cytosolic concentration of ATP that is of 1.3 mM,⁷). HK also requires 5 mM of glucose (typical blood glucose concentration). Whereas it is feasible to add 5 mM glucose in the in vitro assay, the ATP concentration used (1mM) will by itself activate transport (see figure 6). Unfortunately, it is impossible to use lower concentration of ATP as we did for the experiment now shown figure 5b. Indeed, ATP as low as 10 μ M is able to stimulate microtubule-based transport in vitro with significant processive runs of more than 300 nm on MTs⁸. Therefore, this experiment is technically difficult/impossible given the fact that the concentration used to activate HK is higher than the concentration of ATP that activates transport in vitro and, requires extensive set up that was impossible to achieve in the 3 months review process. That is why we focused on demonstrating that purified vesicles can produce ATP from ADP + Pi and glucose with low amounts of ATP as it is now shown in new figure 5b. We hope this will convince reviewer 3 that purified motile vesicles contain functional glycolytic machinery capable of generating ATP from glucose.

Regarding the role of mitochondria in the in vitro assay:

*The P3 fraction that is used for the in vitro assay is devoid of any mitochondria as shown to the reviewer above and now in the revised version as **supplemental figure 1**. Finally, the reviewer should note that the motility buffer used in the in vitro assay is implemented with an oxygene-scavenger system composed by glucose oxidase and catalase, thus creating anoxia in the motility chambers, which would therefore preclude mitochondria respiration even in the hypothesis that functional mitochondria are contaminating the motility vesicular fraction.*

Minor issues:

5. Details on the specific methodology used to track, filter and analyze the axonal and in-vitro motility is severely lacking as this could greatly influence the outcome of the analysis results.

A more detailed description of the way vesicular motility was analyzed was added in the methods section pages 19-20.

6. It should be recognized that as most evidence on the glycolysis-machinery association and function is based on dynactin-enriched vesicular fraction from whole brain areas, these are not necessarily long distance, axon-transported specific compartment. Therefore the authors should refrain from emphasizing their conclusions regarding axonal transport specificity of their findings and model.

We agree with this point especially as we found not only glycolysis on vesicles in axons but also in dendrites (see new Figure 3) and that proteomics was performed on whole brain. This (whole brain & axon specificity and long distance) has been toned down throughout the text (see results and discussion)

7. The authors should also refer to other proteomic studies on axonal transport associated fractions (Michaelovski et al., MCP 2010, Debaisieux et al., MCP 2015, Gershoni-Emek et al., MCP 2015), some of which have shown the association of several glycolytic enzymes in their analysis.

We apologize for not citing these studies and have now included these references in the revised version of the Ms (see results sections page 8)

Overall, this is a very interesting and well-sustained work, providing a novel insight on the mechanisms of neuronal transport and motor based transport in general.

Should the authors address the above comments, I suggest it will be accepted for publication.

REFERENCES

1. Loewy, A., Liu, W.S., Baitinger, C. & Willard, M.B. The major 35S-methionine-labeled rapidly transported protein (superprotein) is identical to SNAP-25, a protein of synaptic terminals. *J Neurosci* **11**, 3412-3421 (1991).
2. Zala, D. *et al.* Vesicular glycolysis provides on-board energy for fast axonal transport. *Cell* **152**, 479-491 (2013).
3. Ahrens, C.H., Brunner, E., Qeli, E., Basler, K. & Aebersold, R. Generating and navigating proteome maps using mass spectrometry. *Nat Rev Mol Cell Biol* **11**, 789-801 (2010).
4. Bantscheff, M., Schirle, M., Sweetman, G., Rick, J. & Kuster, B. Quantitative mass spectrometry in proteomics: a critical review. *Anal Bioanal Chem* **389**, 1017-1031 (2007).
5. Newsholme, E.A., Rolleston, F.S. & Taylor, K. Factors affecting the Glucose 6-Phosphate Inhibition of Hexokinase from Cerebral Cortex Tissue of the Guinea Pig. *Biochem J.* **106**, 193-201 (1968).
6. Mukai, C., Bergkvist, M., Nelson, J.L. & Travis, A.J. Sequential reactions of surface- tethered glycolytic enzymes. *Chem Biol* **16**, 1013-1020 (2009).
7. Rangaraju, V., Calloway, N. & Ryan, T.A. Activity-driven local ATP synthesis is required for synaptic function. *Cell* **156**, 825-835 (2014).
8. Ross, J.L., Wallace, K., Shuman, H., Goldman, Y.E. & Holzbaur, E.L. Processive bidirectional motion of dynein-dynactin complexes in vitro. *Nat Cell Biol* **8**, 562-570 (2006).

REVIEWERS' COMMENTS:

Reviewer #1 (Remarks to the Author):

In my original review I was already quite positive about this manuscript. The authors' reply, which included all of the experiments that I suggested, fully addresses my comments. Therefore, I am happy to suggest that the manuscript be published in its current form.

Reviewer #2 (Remarks to the Author):

All the issues have been dealt with adequately in the revised version of the manuscript. I recommend accepting the paper.

Reviewer #3 (Remarks to the Author):

The authors answer my concerns and the manuscript is ready for publication

Point-by-point answer to reviewers

The three reviewers are positive on the revised version of the manuscript.

REVIEWERS' COMMENTS:

Reviewer #1 (Remarks to the Author):

In my original review I was already quite positive about this manuscript. The authors' reply, which included all of the experiments that I suggested, fully addresses my comments. Therefore, I am happy to suggest that the manuscript be published in its current form.

Reviewer #2 (Remarks to the Author):

All the issues have been dealt with adequately in the revised version of the manuscript. I recommend accepting the paper.

Reviewer #3 (Remarks to the Author):

The authors answer my concerns and the manuscript is ready for publication